# HiPER: Hierarchical Plan–Execute RL for Multi-Turn LLM Agents

**Jiangweizhi Peng** [1]  **Yuanxin Liu** [2]  **Ruida Zhou** [3]  **Charles Fleming** [4]  **Zhaoran Wang** [2]  **Alfredo Garcia** [5]
**Mingyi Hong** [1]

🌐 Project Page    🎧 Code

## Abstract

Training LLMs as interactive agents for multi-turn decision-making remains challenging, particularly in long-horizon tasks with sparse and delayed rewards, where agents must execute extended sequences of actions before receiving meaningful feedback. Most existing reinforcement learning (RL) approaches model LLM agents as flat policies operating at a single time scale, selecting one action at each turn. In sparse-reward settings, such flat policies must propagate credit across the entire trajectory without explicit temporal abstraction, which often leads to unstable optimization and inefficient credit assignment. We propose *HiPER*, a novel Hierarchical Plan–Execute RL framework that explicitly separates high-level planning from low-level execution. HiPER factorizes the policy into a high-level planner that proposes subgoals and a low-level executor that carries them out over multiple action steps. To align optimization with this structure, we introduce a key technique called hierarchical advantage estimation (HAE), which carefully assigns credit at both the planning and execution levels. By aggregating returns over the execution of each subgoal and coordinating updates across the two levels, HAE provides an unbiased gradient estimator and provably reduces variance compared to flat generalized advantage estimation. Empirically, HiPER achieves state-of-the-art performance on challenging interactive benchmarks, reaching 97.4% success on ALFWorld and 83.3% on WebShop with Qwen2.5-7B-Instruct (+6.6% and +8.3% over the best prior method), with especially large gains on long-horizon tasks requiring

multiple dependent subtasks. These results highlight the importance of explicit hierarchical decomposition for scalable RL training of multi-turn LLM agents.

## 1. Introduction

**Motivation.** Large language models (LLMs) are increasingly deployed as agents that must complete tasks through multi-turn interactions with an environment, where effective planning and decision-making are essential for success. Reinforcement learning (RL) has emerged as a dominant paradigm for improving these agentic capabilities. Most existing RL methods model LLM agents as flat policies operating at a single time scale, selecting an action at each turn based on the current observation and interaction history (Schulman et al., 2017; Shao et al., 2024; Wang et al., 2025; Feng et al., 2025; Liu et al., 2025a). While such approaches have led to substantial improvements over pretrained models, a notable performance gap persists on long-horizon tasks with sparse rewards, where agents must execute extended action sequences, which may include over tens of thousands of tokens, before receiving meaningful feedback. In these settings, "flat" RL methods must infer long-range dependencies solely from distant end-of-trajectory signals, often resulting in inefficient credit assignment and unstable behavior (Sutton et al., 1999; Bacon et al., 2017; Nachum et al., 2018; Klissarov et al., 2025).

To better understand this limitation, we inspect successful trajectories of trained flat LLM agents and observe a consistent pattern that has also been widely noted in recent works on agentic LLMs and long-horizon decision making (Wang et al.; Huang et al., 2022; Ahn et al., 2022): long action sequences implicitly organize into segments, each corresponding to an intermediate subgoal that persists over multiple turns. For example, Fig. 1 shows the task "clean some cup and put it in cabinet", which naturally decomposes into locating the cup, cleaning it, and placing it in the cabinet. Such segmentations occur across tasks, with coherent stretches of actions separated by sparse transition points where the agent switches subgoals. This suggests an *implicit hierarchical structure* underlying long-horizon

---
[1]University of Minnesota [2]Northwestern University [3]Amazon AGI [4]Cisco Research [5]Texas A&M University. Correspondence to: Mingyi Hong <mhong@umn.edu>.

*Proceedings of the 43rd International Conference on Machine Learning*, Seoul, South Korea. PMLR 306, 2026. Copyright 2026 by the author(s).

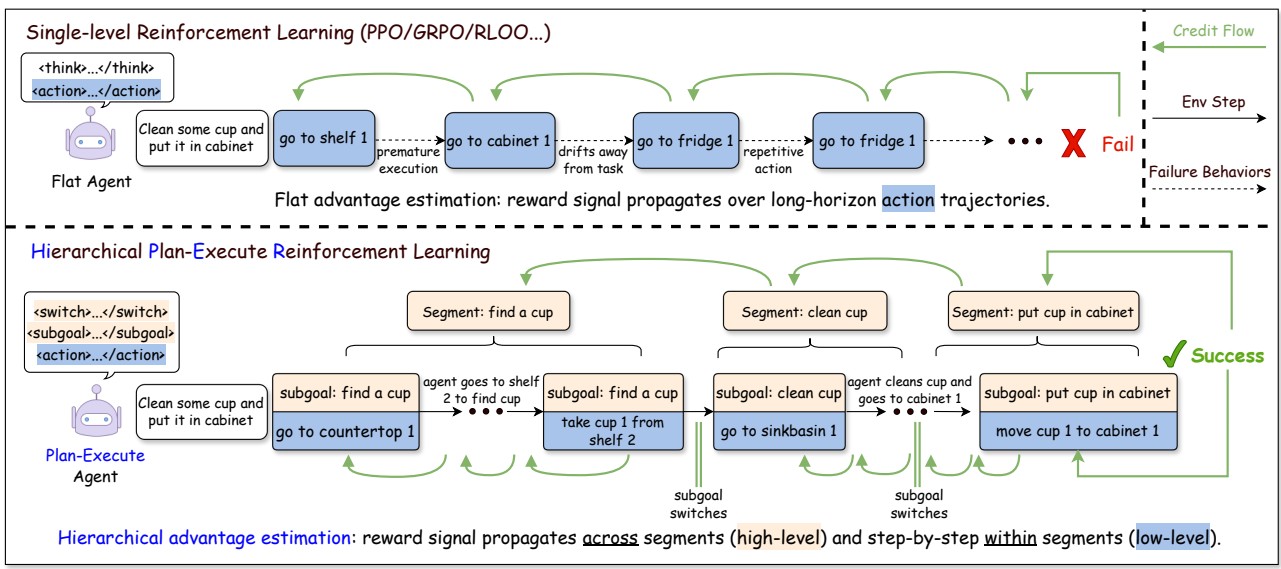

*Figure 1.* **Overview of the HiPER framework.** The upper panel illustrates standard flat RL for LLM agents, where a single policy operates at one time scale and chooses an action at every turn, often leading to brittle long-horizon behavior. For instance, the agent may prematurely head to the cabinet before picking up, and cleaning the cup. The lower panel presents our HiPER framework, built on two components: the Plan-Execute interface (Sec. 4.1), a structured agent interface that explicitly separates high-level *planning* and low-level *execution*; and we propose the hierarchical advantage estimation (Sec. 4.3), which aligns credit assignment with this two-level structure by propagating learning signals both *within* and *across* subgoal segments.

agentic tasks and effective agent behavior, where *temporally extended* subgoals organize low-level actions over multiple turns. While flat RL agents, which typically operate under the ReAct (Yao et al., 2022b) template, may exhibit planning-like behavior within their per-step reasoning, they neither explicitly represent nor optimize for this structure. As a result, any subgoal organization remains *implicit* in the trajectories, often yielding fragile long-horizon behavior, e.g., abandoning unfinished stages or repeatedly taking ineffective actions, as illustrated in Fig. 1.

**The HiPER Framework.** Motivated by these observations, we propose *Hierarchical Plan-Execute Reinforcement learning (HiPER)*, a **hierarchical RL** framework for training LLM agents on long-horizon, sparse-reward tasks. HiPER makes the implicit hierarchical structure in agent behavior explicit by separating slow, high-level planning from fast, low-level action execution, allowing the agent to commit to temporally extended subgoals, decide when to switch between them, and condition action generation on the current subgoal. More specifically, HiPER factorizes the policy into a high-level planner that proposes subgoals and a low-level executor that carries them out over multiple action steps. This structured decomposition introduces intermediate decision points that improve long-horizon coordination and enable more effective credit assignment under sparse feedback. See Fig. 1 for an overview of HiPER.

The proposed HiPER relies on a *Plan–Execute interface*, a structured system prompt template that makes the agent's hierarchical decisions explicit by separating high-level sub-

goal planning from low-level action execution within a *single* auto-regressive LLM policy. At each turn, the agent emits a structured output consisting of (i) a binary switching decision indicating whether to retain or update the current subgoal (i.e., SWITCH or KEEP, enclosed in a `<switch>` block), (ii) the current subgoal description (enclosed in a `<subgoal>` block), and (iii) a primitive environment action (enclosed in the `<action>` block). In this way, the flat agent-environment interaction trajectory is transformed into explicit, learnable planning and execution decisions at two levels. It is important to note that, both the plan and execution in the HiPER framework are *dynamic*. The plans (subgoal generation, subgoal switching) are decided on the fly as the state observations evolve, rather than proposing a subgoal sequence upfront and following the fixed plan throughout. Likewise, the execution is conditioned on the current subgoal, but still determined by the agent step by step, rather than playing fixed sequences of actions pre-specified for each subgoal. This design allows both levels of decisions to be *jointly optimized*, so the agent's abilities of global planning and subtask completion can be simultaneously improved.

Learning under the Plan–Execute framework is non-trivial due to the strong correlation between decisions at different time scales. High-level subgoals shape which low-level actions should be executed over subsequent turns, while the quality of low-level execution determines whether a proposed subgoal is effective. Moreover, a subgoal decision may maintain across multiple turns (depending the quality

of the actions and environment feedback) whereas low-level actions are selected at every turn. This *mismatch* in temporal granularity creates challenges for credit assignment and optimization. To address this, we develop a novel two-time-scale advantage estimation scheme, termed Hierarchical Advantage Estimation (HAE), which provides coupled learning signals for subgoal selection, subgoal switching, and action execution. Intuitively, HAE assigns credit to high-level decisions based on the aggregated outcome of the corresponding subgoal segment, while low-level advantages measure per-step improvement under the current subgoal. This alignment between the learning signal and the hierarchical decision structure enables stable and efficient joint optimization of planning and execution policies.

We provide theoretical justification for the proposed Hierarchical Advantage Estimation (HAE) by establishing **two key properties**. First, we show that HAE yields an unbiased estimator of the policy gradient with respect to both high-level subgoal decisions and low-level action execution, up to standard bootstrapping and value-function approximation errors. Second, we demonstrate that HAE achieves variance reduction compared to flat advantage estimation by aligning credit assignment with the hierarchical structure induced by subgoal segments. Together, these results establish HAE as a principled learning mechanism for jointly optimizing planning and execution in hierarchical LLM agents.

Empirically, we evaluate HiPER on two interactive benchmarks, ALFWorld (Shridhar et al., 2021), a text-based embodied household environment and WebShop (Yao et al., 2022a), a simulated website interaction environment, where it achieves, to our knowledge, state-of-the-art performance, with 97.4% success rate on ALFWorld, and 83.3% success rate on WebShop with Qwen2.5-7B-Instruct.

We summarize our contributions as follows.

• **Implicit hierarchy identification**. We identify a consistent implicit hierarchical structure in successful multi-turn LLM agent behavior and make it explicit through a Plan–Execute interface that separates high-level subgoal planning from low-level action execution.

• **Hierarchical credit assignment.** Central to our HiPER algorithm is HAE, a coupled, two-timescale advantage estimation scheme under the Plan-Execute interface. We provide theoretical guarantees showing unbiasedness of HAE up to approximation errors and provable variance reduction compared to Generalized Advantage Estimation (GAE).

• **Unified hierarchical RL framework.** We integrate the Plan-Execute interface and HAE into the HiPER framework, which has superior empirical performance over baselines on multiple interactive benchmarks.

## 2. Related Work

**Hierarchical RL (HRL).** Classical HRL formalizes temporal abstraction via the *options* framework (Sutton et al., 1999), where a policy selects a temporally extended option with its own intra-option policy and termination, inducing a semi-MDP hierarchy. Early methods assume a fixed option set and execution semantics (Sutton et al., 1998). Later work explores learning options end-to-end (e.g., Option-Critic (Bacon et al., 2017), PPOC (Klissarov et al., 2017), DAC (Zhang & Whiteson, 2019), h-DQN (Kulkarni et al., 2016)) but still fixes the option inventory (e.g., a predetermined number of options). HiPER is motivated by the same temporal abstraction principle, but is not a direct transplant of options to LLM agents: instead of learning a discrete option set, we use open-vocabulary subgoals and introduce a hierarchical advantage estimator that explicitly propagates credit across segment boundaries, yielding stronger learning signals on long-horizon tasks.

**RL for LLM Agents.** Recent work trains LLMs as interactive agents with on-policy RL, typically modeling agents as flat turn/token policies while trying to improve optimization and credit assignment. LOOP adapts PPO with group leave-one-out for long-horizon agent training (Chen et al., 2025); RAGEN (StarPO) studies trajectory-level agent RL and its stability pathologies (Wang et al., 2025); GiGPO uses trajectory- and step-level relative advantages for denser credit signals (Feng et al., 2025); and implicit step rewards from process reward modeling provide additional shaping (Liu et al., 2025b). Our proposed HiPER target the same setting but make hierarchy explicit via Plan–Execute and align advantage estimation with the hierarchy structure.

## 3. Preliminaries

**RL for Interactive LLM agents.** We consider an interactive setting where an LLM-based agent completes multi-step tasks specified by a textual description $x \sim p(X)$. At each environment step $t = 1, 2, \ldots, T$, the agent receives a textual prompt $p_t := \text{Format}(s_t)$, where $s_t \in \mathcal{S}$ is the state observation, $\text{Format}(\cdot)$ is a prompt template that wraps the state in a textual description, and produces a textual action $a_t \in \mathcal{V}^{\leq n}$, i.e., a token sequence over vocabulary $\mathcal{V}$ with maximum length $n$. After executing $a_t$, the environment returns a scalar reward $r_t \in \mathbb{R}$ and transitions to the next state $s_{t+1}$. A full episode is a trajectory $\tau = \{(s_0, a_0, r_0), \ldots, (s_{T-1}, a_{T-1}, r_{T-1})\}$. The objective function of RL training for the agent can be written as:

$$J(\theta) = \mathbb{E}_{x \sim p(X)} \mathbb{E}_{\tau \sim \pi_\theta(\cdot | x)} \left[ \sum_{t=0}^{T-1} \gamma^t r_t \right], \quad (1)$$

where $\theta$ denotes the LLM parameters, and $\gamma \in (0, 1]$ is the discount factor. This formulation treats each interaction

as selecting a primitive action $a_t$ at a single time scale. Next, we introduce the proposed hierarchical formulation for modeling LLM agentic tasks.

**Hierarchical RL for Interactive Agents.** To formally represent the underlying hierarchical structure for LLM agents, we introduce a hierarchical RL formulation based on the options framework (Sutton et al., 1999; Barto & Mahadevan, 2003). Concretely, we model the agent as operating with a high-level option $o_t \in \mathcal{O}$ (e.g., a subgoal) and a low-level action $a_t \in \mathcal{A}$ at each turn. The high-level option $o_t$ in our case corresponds the planning decision, which the agent chooses that guides the low-level actions for executions. These $o_t$'s persists for multiple turns before being switched to a new subgoal, and we call such a period of time a 'planning segment'. The switching time is governed by a binary decision $q_t \in \{0, 1\}$, where $q_t = 1$ indicates terminating the current option and switching to a new one. We parameterize the switching policy by $\eta$, as $q_t \sim \pi_\eta^{\text{switch}}(\cdot \mid s_t, o_{t-1})$. Denote by turn indices $0 = b_0 < b_1 < \cdots < b_K = T$ the *boundary turns*, where termination and switch of option is determined, and $K$ denotes the number of planning segments. At each boundary index $b_k$, the high-level context is updated according to $o_k \sim \pi_\psi^{\text{high}}(\cdot \mid s_{b_k})$. At each turn $t$, the low-level action is generated with $\pi_\phi^{\text{low}}(\cdot \mid s_t, o_t)$. The trajectory resulting from this process can be written as $\tau = \{(s_t, q_t, o_t, a_t, r_t)\}_{t=0}^{T-1}$. The joint hierarchical policy is thus factorized as:

$$
\begin{aligned}
\pi_{\eta,\psi,\phi}(\tau \mid x) = \prod_{t=0}^{T-1} & \underbrace{\pi_\eta(q_t \mid s_t, o_{t-1})}_{\text{switch}} \\
& \times \underbrace{\left( (1-q_t)\mathbf{1}[o_t = o_{t-1}] + q_t \pi_\psi^{\text{high}}(o_t \mid s_t) \right)}_{\text{subgoal}} \\
& \times \underbrace{\pi_\phi^{\text{low}}(a_t \mid s_t, o_t)}_{\text{action}} \underbrace{p(s_{t+1} \mid s_t, a_t)}_{\text{environment}}.
\end{aligned}
\tag{2}
$$

where $\mathbf{1}[\cdot]$ is the indicator function, enforcing that the previous subgoal is deterministically carried over if $q_t = 0$. The RL training objective under this hierarchical factorization can be therefore expressed as:

$$
J(\eta, \psi, \phi) = \mathbb{E}_{x \sim p(X)} \mathbb{E}_{\tau \sim \pi_{\eta,\psi,\phi}(\cdot \mid x)} \left[ \sum_{t=0}^{T-1} \gamma^t r_t \right], \tag{3}
$$

where $\pi_{\eta,\psi,\phi}(\cdot \mid x)$ denotes the hierarchical policy and $\gamma \in (0, 1]$ is the discount factor. The key in the above formulation (2) and (3) is the explicit separation of high-level planning and low-level execution, which provides a structured view of the agent's decisions across time scales and serves as the basis for our Plan–Execute interface and hierarchical learning algorithm introduced in Section 4.

## 4. Method

We adapt the hierarchical RL framework from Section 3 to LLM-based agents, so that the abstract decomposition in (2) is concretely instantiated as a structured auto-regressive generation process, allowing us to leverage the hierarchical structure to design more efficient and stable learning algorithm. In this section, we first introduce the *Plan-Execute agent interface*, a system prompt template designed to instantiate the hierarchical policy defined in (2) for auto-regressive LLMs. This interface explicitly prompts the model to emit planning and execution decisions in a structured format (to be described shortly). Then, we present a novel hierarchical RL algorithm that exploits the hierarchical structure for more efficient and stable learning.

### 4.1. Plan-Execute Framework for Hierarchical RL

We introduce our proposed Plan-Execute framework, in which the LLM agent makes decisions at two time scales: on the high-level, it maintains a persistent subgoal for planning that guides its behavior across multiple turns, as well as decides when to switch to another subgoal; on the low-level, it executes primitive actions that interact with the environment given the current subgoal, as shown in Fig.1. Essentially, we implement the Plan-Execute interface via a structured system prompt template by extending the ReAct (Yao et al., 2022b) prompting. We present complete system prompt templates in Appendix C.3.

At each environment step $t$, the agent receives a textual prompt $p_t = \text{Format}(s_t, o_{t-1})$, containing the current state observation and the previous subgoal, then produces a structured output

$$
\begin{aligned}
\langle q_t, o_t, a_t \rangle = & \texttt{<switch>...</switch>} \\
& \texttt{<subgoal>...</subgoal>} \\
& \texttt{<action>...</action>},
\end{aligned}
\tag{4}
$$

where $q_t$ is a binary switch decision made by the agent, given the state $s_t$ and previous subgoal $o_{t-1}$, enclosed by `<switch>` tags; $o_t \in \mathcal{V}^{\leq m}$ is the (possibly updated) current subgoal text enclosed by `<subgoal>` tags; $a_t \in \mathcal{V}^{\leq n}$ is the primitive action with `<action>` tags. Particularly, we only allow $q_t$ to be either `<switch>SWITCH</switch>` or `<switch>KEEP</switch>`. For simplicity, we write $q_t = 1$ if the decision is SWITCH and $q_t = 0$ if KEEP. When $q_t = 0$, the agent retains the previous subgoal and copies as-is to the current subgoal, i.e., $o_t = o_{t-1}$; when $q_t = 1$, it generates a new subgoal $o_t$.

Using the above template, our proposed Plan-Execute framework induces the three policies specified in (2). Formally, let $\theta$ be the LLM parameter, at step $t$ we first sample the switch decision $q_t \sim \pi_\theta(\cdot \mid s_t, o_{t-1})$. If $q_t = 1$ we sample

a new subgoal $o_t \sim \pi_\theta(\cdot \mid s_t)$, otherwise $o_t = o_{t-1}$. It then follows by a low-level action conditioned on the current option, $a_t \sim \pi_\theta(\cdot \mid s_t, o_t)$. The switching, subgoal, and action policies $\pi_\eta^{\text{switch}}$, $\pi_\psi^{\text{high}}$, and $\pi_\phi^{\text{low}}$ are realized by a **single auto-regressive LLM policy** $\pi_\theta$ that emits $(q_t, o_t, a_t)$. Importantly, the switching, subgoal and action fields are generated *in order*, so the correct conditioning is directly achieved through the LLM's auto-regressive factorization.

By directly leveraging the LLM's native auto-regressive factorization without introducing separate high- or low-level controllers, this Plan-Execute instantiation yields a convenient policy decomposition that simplifies subsequent computation and algorithm design. However, specifying the Plan-Execute structure alone is not sufficient. Plan-Execute defines what decisions (switch/subgoal/action) are generated and when, but it remains unclear how these hierarchical, coupled decisions could be effectively learned, especially under long-horizon, sparse reward settings. To this end, we next derive the policy gradient under the Plan-Execute factorization. This derivation reveals that the learning signal naturally decomposes into level-specific terms corresponding to switching, subgoal selection, and low-level actions, which in turn motivates hierarchical advantage estimation and more stable updates that explicitly exploit the policy structure.

### 4.2. Plan-Execute Policy Gradient

Given the Plan-Execute policy factorization in (2) and our learning objective defined in (3), the following theorem decomposes the policy gradient into contributions from switching, subgoal, and action decisions.

**Theorem 4.1** (Plan-Execute Gradient). *Assume the Plan-Execute policy is given by the conditionals $\pi_\theta(q_t \mid s_t, o_{t-1})$, $\pi_\theta(o_t \mid s_t)$ (invoked only when $q_t = 1$), and $\pi_\theta(a_t \mid s_t, o_t)$. Then the gradient of (3) is*

$$
\nabla_\theta J(\theta) = \mathbb{E}_{x \sim p(X)} \mathbb{E}_{\tau \sim \pi_\theta} \Bigg[ \sum_{t=0}^{T-1} \Big( \nabla_\theta \log \pi_\theta(q_t \mid s_t, o_{t-1}) A_t^{\text{switch}}
$$
$$
+ q_t \nabla_\theta \log \pi_\theta(o_t \mid s_t) A_t^{\text{high}}
$$
$$
+ \nabla_\theta \log \pi_\theta(a_t \mid s_t, o_t) A_t^{\text{low}} \Big) \Bigg],
$$

(5)

*where the advantages are defined by:*

$$
A_t^{\text{switch}} := \underbrace{\mathbb{E}[G_t \mid s_t, o_{t-1}, q_t]}_{Q^{\text{switch}}(s_t, o_{t-1}, q_t)} - \underbrace{\mathbb{E}[G_t \mid s_t, o_{t-1}]}_{V^{\text{switch}}(s_t, o_{t-1})},
$$
$$
A_t^{\text{high}} := \underbrace{\mathbb{E}[G_t \mid s_t, q_t = 1, o_t]}_{Q^{\text{high}}(s_t, o_t)} - \underbrace{\mathbb{E}[G_t \mid s_t, q_t = 1]}_{V^{\text{high}}(s_t)},
$$
$$
A_t^{\text{low}} := \underbrace{\mathbb{E}[G_t \mid s_t, o_t, a_t]}_{Q^{\text{low}}(s_t, o_t, a_t)} - \underbrace{\mathbb{E}[G_t \mid s_t, o_t]}_{V^{\text{low}}(s_t, o_t)},
$$

$G_t := \sum_{t'=t}^{T-1} \gamma^{t'-t} r_{t'}$ *is the return-to-go, $T$ denotes the total*

*number of environment steps, and expectations are taken over future rollout from time $t$ onward.*

Proof of Theorem 4.1 is deferred to Appendix A.1. Theorem 4.1 indicates that the gradient of Plan-Execute policy naturally decomposes into coupled components operating at two different time scales: a high-level process that optimizes subgoal selection and switching through $A_t^{\text{high}}$ and $A_t^{\text{switch}}$, and a low-level process that optimizes primitive action generation within each subgoal segment through $A_t^{\text{low}}$. Next, we discuss how to properly estimate the advantages $A_t^{\text{switch}}, A_t^{\text{high}}, A_t^{\text{low}}$ from the rollout trajectories.

### 4.3. Hierarchical Advantage Estimation

Theorem 4.1 clarifies how the Plan–Execute factorization decomposes the learning signal across switching, subgoal, and action decisions. We now turn this decomposition into a practical credit-assignment scheme by constructing advantage estimators tailored to each level. Given a batch of on-policy trajectories, our goal is to construct advantage estimators as reliable learning signals for subgoal switching, subgoal generation, and action execution, that facilitate efficient and stable policy optimization, under potentially sparse and delayed rewards. A direct approach is to use Monte-Carlo returns, combined with group baselines, such as in GRPO (Shao et al., 2024) and RLOO (Ahmadian et al., 2024), to construct step-level advantages. While doing so avoids training a critic model, they require sampling multiple rollouts per decision point, which becomes prohibitively expensive in interactive long-horizon environments (Feng et al., 2025). Therefore, we adopt an on-policy, PPO-style actor-critic update with learned value baselines.

We propose the Hierarchical Advantage Estimation (HAE), built upon Generalized Advantage Estimation (GAE) (Schulman et al., 2015), and extend it to the Plan–Execute structure by constructing *hierarchical* advantage estimates at two time scales. The challenge of this extension involves (i) aligning learning signals across time scales, and (ii) handling the inherent coupling between levels. While prior option-critic-based methods (Bacon et al., 2017; Klissarov et al., 2017; Zhang & Whiteson, 2019) construct and optimize hierarchical policies similar to (2), they typically estimate high- and low-level learning signals with largely parallel targets, leaving the cross-level coupling unaddressed. We tackle these challenges via a novel boundary-aware bootstrapping and critic learning that judiciously couples the two levels. Concretely, the proposed HAE learns value baselines for low-level action execution and high-level subgoal decisions *jointly*, which are used to define hierarchical TD residuals and GAE-style advantages. We next describe how we derive low-level (action) and high-level (subgoal generation and switching) advantages from a sampled trajectory.

First, the SWITCH decisions partition the trajectory into seg-

ments in which the subgoal remains constant. Within each segment, we compute turn-level TD residuals and estimate advantages for action execution, conditioned on the current option. In parallel, we aggregate rewards within each segment and form segment-level TD residuals at switching boundaries, yielding a GAE-style advantage for subgoal selection. Notably, the turn-level GAE is computed *within* each segment, but the final-step residual in each segment bootstraps to the segment-level value at the next option boundary, which couples within-segment credit assignment with boundary-to-boundary progress. Finally, we derive a switching advantage for the SWITCH/KEEP decisions, capturing the incremental benefit of terminating the current option versus continuing it at the same state.

More formally, recall $0 = b_0 < b_1 < \cdots < b_K = T$ are the switching boundary indices, where $T$ is the episode length, and let the $k$-th segment be the time interval $t \in [b_k, b_{k+1} - 1]$ during which subgoal $o_k$ is active. Let $G_t := \sum_{t'=t}^{T-1} \gamma^{t'-t} r_{t'}$ denote the return-to-go, we define the high-level value baseline $V^{\text{high}}(s_t) := \mathbb{E}[G_t \mid s_t, q_t = 1]$ and the low-level value baseline $V^{\text{low}}(s_t, o_t) := \mathbb{E}[G_t \mid s_t, o_t]$, which are used in the following advantage estimation. Intuitively, the high-level value models the expected return at decision points where the agent proposes a *new* subgoal, serving as the baseline for the advantage estimation of subgoal selection at segment boundaries; the low-level value models the expected return given the *current* subgoal commitment, serving as the baseline for the advantage estimation of action selection within the segment.

**Execution advantage (low-level for action selection).** We apply a GAE-style estimator restricted to each segment: for $t \in [b_k, b_{k+1} - 1]$, the low-level TD residual and the corresponding advantage are:

$$
\begin{aligned}
\delta_t^{\text{low}} &= r_t + \gamma V_t^{\text{next}} - V^{\text{low}}(s_t, o_k), \\
\hat{A}_t^{\text{low}} &= \sum_{\ell=t}^{b_{k+1}-1} (\gamma \lambda^{\text{low}})^{\ell-t} \delta_\ell^{\text{low}}, \quad t \in [b_k, b_{k+1} - 1],
\end{aligned}
\tag{6}
$$

where

$$
V_t^{\text{next}} = \begin{cases} V^{\text{high}}(s_{b_{k+1}}), & t = b_{k+1} - 1 \\ V^{\text{low}}(s_{t+1}, o_k), & \text{otherwise.} \end{cases}
\tag{7}
$$

Here, $V^{\text{low}}$ represents the low-level value of state $s$ conditioned on subgoal $o$, $V^{\text{high}}$ represents the high-level value at boundary states, and $\lambda^{\text{low}}$ is the low-level TD parameter. This assigns fine-grained credit to primitive actions inside each segment. Notably, as reflected in (7), the low-level learning is connected to the high-level process by bootstrapping the final-step residual to the segment-level value at the next subgoal boundary.

**Planning advantage (high-level for subgoal generation).** Each segment is compressed into a macro-step, with segment-level reward $\tilde{r}_k = \sum_{t=b_k}^{b_{k+1}-1} \gamma^{t-b_k} r_t$ and duration

discount $\tilde{\gamma}_k = \gamma^{b_{k+1}-b_k}$. The high-level TD residual and advantages are:

$$
\begin{aligned}
\delta_k^{\text{high}} &= \tilde{r}_k + \tilde{\gamma}_k V^{\text{high}}(s_{b_{k+1}}) - V^{\text{high}}(s_{b_k}), \\
\hat{A}_{b_k}^{\text{high}} &= \sum_{j=k}^{K-1} \left( \prod_{i=k}^{j-1} \tilde{\gamma}_i \lambda^{\text{high}} \right) \delta_j^{\text{high}}, \quad k \in [0, K-1],
\end{aligned}
\tag{8}
$$

where $V^{\text{high}}(s)$ represents the high-level value of state $s$, and $\lambda^{\text{high}}$ is the high-level TD parameter.

**Switching advantage (high-level for subgoal switching).** The switching decision is a *binary* choice between committing to the current subgoal versus terminating and handing control back to the high-level plan. We therefore first define a per-state switching gain,

$$
\delta_t^{\text{switch}} := V^{\text{high}}(s_t) - V^{\text{low}}(s_t, o_{t-1}),
\tag{9}
$$

which measures how much better it is, in expectation, to switch to a new subgoal, rather than continue executing the previous subgoal from the same state. Define $\beta_t := \pi_\theta(q_t = 1 \mid s_t, o_{t-1})$. The learning signal for the realized binary decision is

$$
\hat{A}_t^{\text{switch}} = (q_t - \beta_t) \delta_t^{\text{switch}},
\tag{10}
$$

which can be interpreted as a centered policy-gradient estimator for the binary choice: it increases the log-probability of switching when $q_t = 1$ and switching is estimated to be beneficial ($\delta_t^{\text{switch}} > 0$), and decreases it when switching is taken but is suboptimal ($\delta_t^{\text{switch}} < 0$). Once the advantages $\hat{A}^{\text{high}}$, $\hat{A}^{\text{low}}$, and $\hat{A}^{\text{switch}}$ are calculated, they can be readily plugged into the gradient in (5) for policy updates.

**Critic Learning.** Our hierarchical advantage estimators rely on two value baselines: a low-level critic $V^{\text{low}}(s_t, o_k)$ for action execution within a segment, conditioned on the current subgoal, and a high-level critic $V^{\text{high}}(s_{b_k})$ for subgoal decisions at switching boundaries. In practice, we do not have access to the exact value baselines, therefore, we train critic models to fit the values at both high- and low-level. Let $\phi$ denote the critic model parameters, $V_\phi^{\text{low}}(s, o)$ and $V_\phi^{\text{high}}(s)$ denote the low-level and high-level critic, respectively.

Concretely, we train the high-level critic $V_\phi^{\text{high}}$ to regress to the segment-level bootstrapped target $y^{\text{high}}$:

$$
y_k^{\text{high}} := \tilde{r}_k + \tilde{\gamma}_k \text{sg}(V_\phi^{\text{high}}(s_{b_{k+1}})), \quad k \in [0, K-1],
\tag{11}
$$

where $\text{sg}(\cdot)$ denotes stop gradient. Then the loss function for the high-level critic is:

$$
\mathcal{L}_{V^{\text{high}}}(\phi) := \mathbb{E}\left[ \sum_{k=0}^{K-1} \left( V_\phi^{\text{high}}(s_{b_k}) - y_k^{\text{high}} \right)^2 \right],
\tag{12}
$$

Similarly, we train the low-level critic $V_\phi^{\text{low}}$ to regress to the bootstrapped turn-level target $y^{\text{low}}$:

$$y_t^{\text{low}} = r_t + \gamma\,\text{sg}(\hat{V}_t^{\text{next}}), \qquad t \in [b_k, b_{k+1} - 1], \quad (13)$$

where $\hat{V}^{\text{next}}$ is defined as:

$$\hat{V}_t^{\text{next}} = \begin{cases} V_\phi^{\text{high}}(s_{b_{k+1}}), & t = b_{k+1} - 1 \\ V_\phi^{\text{low}}(s_{t+1}, o_k), & \text{otherwise.} \end{cases} \quad (14)$$

Then the loss for the low-level critic is:

$$\mathcal{L}_{V^{\text{low}}}(\phi) := \mathbb{E}\left[ \sum_{t=0}^{T-1} \left( V_\phi^{\text{low}}(s_t, o_k) - y_t^{\text{low}} \right)^2 \right] \quad (15)$$

Notably, the low-level and high-level critic learning are coupled in the same way as in advantage estimation, so that the final-step target in each segment bootstraps to the boundary high-level value $V^{\text{high}}(s_{b_{k+1}})$, ensuring that the low-level value estimates remain consistent with high-level boundary returns and can propagate learning signals across segments. Importantly, although HAE involves two value functions, it does not require training two separate critics. In practice, we use a single shared critic backbone with two output heads, only incurring *negligible* memory overhead relative to standard PPO (see Appendix C.2).

We optimize the Plan-Execute actor and critic models using the PPO-style clipped objectives, as defined in Equations (28), (29) and (30) in Appendix B, with the advantages and critic targets discussed above. We summarize the complete hierarchical learning algorithm in Algorithm 1, which is deferred to Appendix B, along with full PPO loss definitions and implementation details. In short, the full training loop of HiPER described in Algorithm 1 involves collecting Plan-Execute rollouts, estimating hierarchical advantages from the rollouts, calculating the actor and critic losses, and updating parameters via PPO-style clipped objectives with KL regularization.

### 4.4. Unbiasedness and Variance Reduction

In this section, we establish two theoretical properties of the policy-gradient estimator induced by our hierarchical advantage estimation: (i) it is *unbiased* up to the GAE bootstrapping and critic approximation errors, and (ii) it achieves *variance reduction* relative to a flat GAE baseline by exploiting the hierarchical structure of Plan-Execute trajectories.

**Theorem 4.2.** *Let $\hat{g}(\theta)$ be the gradient estimate by substituting $\{\hat{A}_t^{switch}, \hat{A}_{b_k}^{high}, \hat{A}_t^{low}\}$ from (6), (8) and (10) into (5), with learned critics $\hat{V}^{low}$ and $\hat{V}^{high}$, into the gradient in (5). Define the corresponding oracle estimator $g^\lambda(\theta)$ by using the same formulas but with the true value functions $V^{low}, V^{high}$. Then*

$$\mathbb{E}\big[\hat{g}(\theta)\big] - \nabla_\theta J(\theta) = \underbrace{\Big(\mathbb{E}\big[g^\lambda(\theta)\big] - \nabla_\theta J(\theta)\Big)}_{\textit{GAE bootstrapping bias}} + \underbrace{\mathbb{E}\big[\hat{g}(\theta) - g^\lambda(\theta)\big]}_{\textit{critic approximation bias}}.$$

*In particular, when $\lambda^{low} = \lambda^{high} = 1$ (Monte Carlo estimation) and $\hat{V}^{low} = V^{low}, \hat{V}^{high} = V^{high}$ (critics are perfectly learned), $\mathbb{E}[\hat{g}(\theta)] = \nabla_\theta J(\theta)$, i.e. $\hat{g}(\theta)$ is an unbiased stochastic gradient estimator.*

**Theorem 4.3** (Informal). *Consider the same Plan–Execute policy $\pi_\theta$ and on-policy rollouts. Let $A_t^{flat}$ denote the advantage obtained from applying standard flat GAE to the Plan–Execute trajectory, and let $A_t^{HAE}$ denote the low-level execution advantage produced by HAE. Under simplifying assumptions (e.g., exact value baselines, $\lambda^{low} = \lambda^{high} = 1$),*

$$Var\big(A_t^{HAE}\big) \leq Var\big(A_t^{flat}\big),$$

*with strict inequality whenever subgoals and switching boundaries carry nontrivial information about future returns beyond the state.*

Proof of Theorems 4.2 and 4.3, as well as the formal Theorem 4.3 are deferred to Appendix A.

## 5. Experiments

In this section, we present the empirical evaluation of our method on two agentic tasks: ALFWorld (Shridhar et al., 2021) and WebShop (Yao et al., 2022a). Compared to baselines, our HiPER framework demonstrates: (i) superior final task performance in long-horizon, sparse-reward settings; (ii) substantially improved sample efficiency and training stability; and (iii) clear and interpretable subgoals that better structure the agent's behavior during execution.

### 5.1. Experimental Setup

**Tasks.** We train the HiPER agent on two challenging tasks: ALFWorld and WebShop. ALFWorld is an interactive TextWorld environment that generates textual descriptions of the physical world and responds to textual actions by the agent. The agent's task is to complete a given household activity through textual interaction with the environment. ALFWorld contains six task categories of different complexities: Pick & Place (Pick), Examine in Light (Look), Clean & Place (Clean), Heat & Place (Heat), Cool & Place (Cool), Pick Two & Place (Pick2). WebShop is an interactive web-based environment that emulates the task of online shopping on an e-commerce website. The goal is to understand the given text instruction and purchase a product to match the user's specifications by interacting with the simulated website.

**Baselines.** We compare HiPER against the following baselines: (i) PPO (Schulman et al., 2017), the standard actor-critic RL method widely used in RLHF and LLM agent training; (ii) RLOO (Ahmadian et al., 2024), a critic-free RL method with group-based value baseline; (iii) GRPO (Shao et al., 2024), which forms trajectory-level advantage via group-based reward normalization; and (iv) GiGPO (Feng

et al., 2025), a recent agentic RL method that assigns step-level advantages using state-wise grouping and has shown strong empirical performance. As reference, we also report the performance of the base model before RL training.

**Evaluation.** We use Qwen2.5-1.5B-Instruct and Qwen2.5-7B-Instruct (Team, 2024) as base models for RL training. All evaluated RL methods use the same set of hyperparameters provided in Appendix C.1. We adopt the ReAct (Yao et al., 2022b) prompt template for all the baseline methods unless otherwise stated, and use our proposed Plan-Execute prompt template for HiPER. Both prompt templates are provided in Appendix C.3. For ALFWorld, we evaluate the success rate across all six task categories and the overall success rate. For WebShop, we evaluate the agent's task score and success rate. For both tasks we set the total training epochs to be 150, following GiGPO, to ensure fair comparison.

### 5.2. Experimental Results

Table 1 presents the main results on ALFWorld and WebShop tasks. We summarize key takeaways below.

**RL training substantially improve model performance.** For both 1.5B and 7B models, PPO, RLOO, and GRPO substantially improve performance over the base model across both benchmarks, underscoring the importance of RL training for multi-turn interactive tasks. With the 1.5B model, PPO increases ALFWorld overall success from 8.3% to 68.2% and WebShop success from 5.5% to 51.5% (score 73.8). Similar trends hold for the 7B model, where PPO reaches 82.8% overall success on ALFWorld and 68.7% on WebShop; RLOO and GRPO yield similar overall gains. GiGPO further improves performance by leveraging denser step-level signals in addition to the episode outcome, achieving 90.8% on ALFWorld and 75.2% on WebShop.

**Baselines struggle most on tasks requiring multiple sequential subtasks.** A consistent pattern in Table 1 is that standard RL baselines lag most on ALFWorld task categories whose success require *multiple sequential subtasks*. For example, in Pick Two & Place (Pick2) task, the agent must retrieve two target objects sequentially and place them in the correct location; and in Examine in Light (Look) task, the agent needs to pick up a desired object, find and turn on a light source and examine the object in a *sequential* manner. For 1.5B model, PPO/RLOO/GRPO achieve less than 60% success rate on Pick2 and Look, as opposed to more than 70% on simpler tasks such as Pick. Similar pattern can be observed for the stronger baseline GiGPO and on the 7B model as well. These results strongly suggest that single-timescale "flat" RL optimization is less reliable when tasks are composed of several dependent subgoals.

**HiPER achieves the best overall performance, especially**

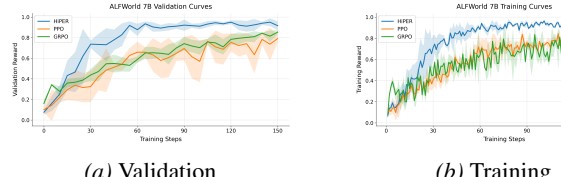

*(a)* Validation       *(b)* Training

*Figure 2.* **ALFWorld 7B Curves.** From the validation curve, HiPER achieves roughly 2.8× speedup relative to PPO/GRPO. From the training curve, HiPER exhibits more stable training dynamics compared with PPO/GRPO, showing smaller oscillations.

**for challenging tasks.** HiPER consistently delivers the strongest performance across both ALFWorld and WebShop tasks, for both model sizes. With Qwen2.5-1.5B, HiPER reaches 95.3% overall success on ALFWorld and 71.4% success on WebShop, outperforming the strongest reported baseline GiGPO by +8.6% and +4.0%, respectively. With Qwen2.5-7B, HiPER further improves to 97.4% on ALFWorld and 83.3% on WebShop, both exceeding GiGPO by around +7%. Notably, the largest gains on ALFWorld come from the more challenging categories such as Look and Pick2, highlighting the clear advantage of explicitly decomposing complex multi-stage tasks into subgoals and learning with a hierarchical structure.

### 5.3. Results Analysis

Fig. 2a compares validation success on ALFWorld over training steps for HiPER, PPO, and GRPO. Across both model sizes, HiPER improves faster and converges higher than the flat baselines. For the 7B model, PPO/GRPO take roughly 140 steps to reach ~80% success, while HiPER exceeds 80% in about 50 steps, a 2.8× sample-efficiency gain. A similar trend holds at 1.5B (Appendix D), where HiPER reaches the high-success regime earlier, yielding a 2.5× speedup. Besides faster convergence, HiPER also exhibits more stable learning dynamics throughout training. Fig. 2b shows the training dynamics of HiPER, GRPO averaged over 3 seeds. Across the whole training process, HiPER shows smaller oscillations and fewer sharp regressions in the learning trajectory, especially compared with the critic-free GRPO. This suggests that HiPER's updates are more reliable in long-horizon settings, where flat baselines are more prone to noisy learning dynamics.

**Subgoal generation and switching behavior.** Although HiPER receives no external supervision on subgoals, it learns to generate meaningful subgoals that structure behavior and support task completion. We present representative trajectories of HiPER agents in Appendix E. In addition, Fig. 3 shows the switching diagnostics of HiPER during training in ALFWorld. Overall, it exhibits a two-phase learning process: an initial exploratory phase, where switching happens frequently, followed by a consolidation

*Table 1.* Performance on ALFWorld and WebShop. We report the success rate (%) of all six task categories, and the overall success rate for ALFWorld. For WebShop, we report the task score and success rate (%). Results are averaged over 3 random seeds. † indicates numbers reported in the GiGPO paper ([Feng et al., 2025](#)). The best values are in **bold**, and second best values are underlined.

| Method | ALFWorld | | | | | | | WebShop | |
|---|---|---|---|---|---|---|---|---|---|
| | Pick | Look | Clean | Heat | Cool | Pick2 | All | Score | Succ. |
| *Qwen2.5-1.5B-Instruct* | | | | | | | | | |
| Base Model | 15.9 $_{\pm1.8}$ | 13.9 $_{\pm6.7}$ | 11.2 $_{\pm4.1}$ | 3.5 $_{\pm6.1}$ | 0.0 $_{\pm0.0}$ | 4.2 $_{\pm0.7}$ | 8.3 $_{\pm0.9}$ | 25.1 $_{\pm8.9}$ | 5.5 $_{\pm6.3}$ |
| +PPO | 74.0 $_{\pm9.0}$ | 37.5 $_{\pm21.7}$ | 67.0 $_{\pm5.1}$ | 85.6 $_{\pm17.1}$ | 68.8 $_{\pm3.7}$ | 56.1 $_{\pm12.2}$ | 68.2 $_{\pm1.8}$ | 73.8$^{\dagger}_{\pm3.0}$ | 51.5$^{\dagger}_{\pm2.9}$ |
| +RLOO† | 88.3 $_{\pm3.0}$ | 52.8 $_{\pm8.6}$ | 71.0 $_{\pm5.9}$ | 62.8 $_{\pm8.7}$ | 66.4 $_{\pm5.5}$ | 56.9 $_{\pm4.7}$ | 69.7 $_{\pm2.5}$ | 73.9 $_{\pm5.6}$ | 52.1 $_{\pm6.7}$ |
| +GRPO | 77.4 $_{\pm6.7}$ | 54.2 $_{\pm7.2}$ | 75.6 $_{\pm8.6}$ | 85.6 $_{\pm17.1}$ | 67.8 $_{\pm7.4}$ | 56.1 $_{\pm11.0}$ | 71.1 $_{\pm8.2}$ | 75.8$^{\dagger}_{\pm3.5}$ | 56.8$^{\dagger}_{\pm3.8}$ |
| +GiGPO† | 94.4 $_{\pm5.9}$ | 67.5 $_{\pm4.6}$ | 94.8 $_{\pm3.8}$ | **94.4** $_{\pm7.8}$ | 79.8 $_{\pm4.7}$ | 76.4 $_{\pm5.4}$ | 86.7 $_{\pm1.7}$ | 83.5 $_{\pm1.8}$ | 67.4 $_{\pm4.5}$ |
| +HiPER | **98.9** $_{\pm1.9}$ | **91.7** $_{\pm14.4}$ | **97.5** $_{\pm4.3}$ | 90.9 $_{\pm4.7}$ | **96.7** $_{\pm2.8}$ | **91.3** $_{\pm9.6}$ | **95.3** $_{\pm1.4}$ | **85.7** $_{\pm3.2}$ | **71.4** $_{\pm9.0}$ |
| *Qwen2.5-7B-Instruct* | | | | | | | | | |
| Base Model | 27.6 $_{\pm13.2}$ | 26.4 $_{\pm13.2}$ | 17.5 $_{\pm1.3}$ | 0.0 $_{\pm0.0}$ | 5.4 $_{\pm1.8}$ | 4.2 $_{\pm0.3}$ | 14.1 $_{\pm4.1}$ | 46.2$^{\dagger}$ | 19.5$^{\dagger}$ |
| +PPO | 98.0 $_{\pm2.8}$ | 68.8 $_{\pm8.8}$ | 82.5 $_{\pm3.5}$ | 95.0 $_{\pm7.1}$ | 52.5 $_{\pm10.6}$ | 75.0 $_{\pm0.0}$ | 82.8 $_{\pm1.1}$ | 81.4$^{\dagger}_{\pm3.1}$ | 68.7$^{\dagger}_{\pm5.1}$ |
| +RLOO† | 87.6 $_{\pm4.3}$ | 78.2 $_{\pm8.3}$ | 87.3 $_{\pm5.8}$ | 81.3 $_{\pm7.6}$ | 71.9 $_{\pm5.2}$ | 48.9 $_{\pm8.4}$ | 75.5 $_{\pm4.6}$ | 80.3 $_{\pm3.2}$ | 65.7 $_{\pm4.0}$ |
| +GRPO | 97.2 $_{\pm2.9}$ | 68.4 $_{\pm5.9}$ | 86.4 $_{\pm6.9}$ | 81.1 $_{\pm10.2}$ | 84.1 $_{\pm4.0}$ | 75.9 $_{\pm8.5}$ | 85.4 $_{\pm2.0}$ | 79.3$^{\dagger}_{\pm2.8}$ | 66.1$^{\dagger}_{\pm3.7}$ |
| +GiGPO† | 97.7 $_{\pm1.6}$ | 82.7 $_{\pm7.9}$ | 98.8 $_{\pm1.6}$ | 83.7 $_{\pm7.2}$ | 89.3 $_{\pm8.2}$ | 79.2 $_{\pm6.6}$ | 90.8 $_{\pm1.3}$ | 86.2 $_{\pm2.6}$ | 75.2 $_{\pm3.8}$ |
| +HiPER | **100** $_{\pm0.0}$ | **84.8** $_{\pm2.6}$ | **100** $_{\pm0.0}$ | **96.3** $_{\pm6.4}$ | **100.0** $_{\pm0.0}$ | **95.5** $_{\pm4.4}$ | **97.4** $_{\pm1.6}$ | **92.2** $_{\pm0.3}$ | **83.3** $_{\pm0.9}$ |

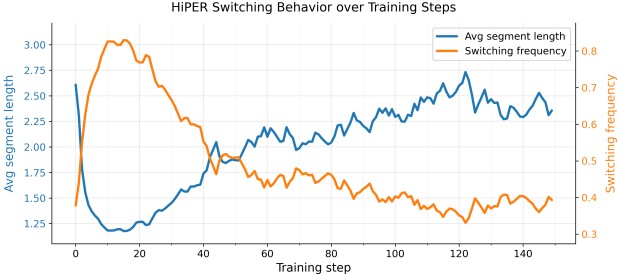

*Figure 3.* **HiPER Switching Behavior on ALFWorld.** The switching frequency increases during early training, indicating a high-level exploration phase. After initial exploration, the switching frequency and segment length stabilizes.

phase where the agent gradually learns to commit to a sub-goal for multiple steps and switching when needed. These observations suggest that HiPER can acquire useful and meaningful high-level planning behavior from outcome-only rewards, rather than collapsing to degenerate strategies such as switching at every turn or no subgoal switches at all.

### 5.4. Ablation on Plan-Execute

To isolate the effects of Plan-Execute and HAE, we investigate the training performance of baseline methods with Plan-Execute prompting. Specifically, during RL training and evaluation, we only change the system prompts of PPO, GRPO, and GiGPO from ReAct to Plan-Execute, and evaluate these methods on ALFWorld. From Table 2 in Appendix D.1, applying Plan-Execute on the base model without RL training harms the model performance, reducing

the initial overall success rate from 8.3% to 2.9%, possibly due to the base model's limited ability to follow instructions. However, training the model with Plan-Execute improves the final performance in general. We observe considerable improvements from PPO with ReAct to PPO with Plan-Execute (+13.1%), and GiGPO with ReAct to GiGPO Plan-Execute (+4.4%). In addition, HiPER remain the best method overall, with a +4.2% advantage over GiGPO with Plan-Execute.

## 6. Conclusion

We introduce HiPER, a novel hierarchical RL framework for training LLM agents. HiPER explicitly separate high-level planning and low-level execution via a Plan-Execute interface, and optimizes its hierarchical policy with a matching hierarchical advantage estimator. By coupling within-segment credit assignment with boundary-to-boundary progress signals, HiPER delivers more stable learning and higher success rates on interactive benchmarks. These results suggest that explicitly modeling and optimizing the multi-timescale structure of agent behavior is a key ingredient for scaling RL to reliably train LLM agents on truly long-horizon tasks with sparse feedback.

## Acknowledgements

We thank the anonymous reviewers for their insightful comments that helped improve this paper. We appreciate support from the following sources: M. Hong and J. Peng are supported in part by NSF grants IIS-2435820 and CIF-2414372, as well as a Cisco Research Award. M. Hong and A. Garcia

are supported in part by NSF grant CIF-2414373.

## Impact Statement

This work improves the reliability and efficiency of reinforcement learning for LLM-based agents in long-horizon, sparse-reward environments by introducing hierarchy-aware credit assignment aligned with plan–execute structure. By reducing common failure modes such as premature stage abandonment and repetitive actions, it can enable more dependable task completion in interactive settings (e.g., household, web, and tool-use). More capable long-horizon agents may also increase risks of misuse or unintended behavior; these concerns highlight the importance of pairing improved training methods with safety measures such as constrained action spaces, environment-level safeguards, monitoring, and careful evaluation prior to deployment.

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

# A. Proofs

## A.1. Proof of Theorem 4.1

**Theorem A.1.** *Assume the Plan-Execute policy is given by the conditionals* $\pi_\theta(q_t \mid s_t, o_{t-1})$, $\pi_\theta(o_t \mid s_t)$ *(invoked only when* $q_t = 1$*), and* $\pi_\theta(a_t \mid s_t, o_t)$. *Let* $G_t := \sum_{t'}^{T-1} \gamma^{t'-t} r_{t'}$ *denote the return-to-go, then the gradient of* (3) *is*

$$\nabla_\theta J(\theta) = \mathbb{E}_{x \sim p(X)} \mathbb{E}_{\tau \sim \pi_\theta} \left[ \sum_{t=0}^{T-1} \left( \nabla_\theta \log \pi_\theta(q_t \mid s_t, o_{t-1}) A_t^{\text{switch}} + q_t \nabla_\theta \log \pi_\theta(o_t \mid s_t) A_t^{\text{high}} + \nabla_\theta \log \pi_\theta(a_t \mid s_t, o_t) A_t^{\text{low}} \right) \right],$$
(16)

*where the advantages are defined by:*

$$A_t^{\text{switch}} := \underbrace{\mathbb{E}[G_t \mid s_t, o_{t-1}, q_t]}_{Q^{\text{switch}}(s_t, o_{t-1}, q_t)} - \underbrace{\mathbb{E}[G_t \mid s_t, o_{t-1}]}_{V^{\text{switch}}(s_t, o_{t-1})},$$

$$A_t^{\text{high}} := \underbrace{\mathbb{E}[G_t \mid s_t, q_t = 1, o_t]}_{Q^{\text{high}}(s_t, o_t)} - \underbrace{\mathbb{E}[G_t \mid s_t, q_t = 1]}_{V^{\text{high}}(s_t)},$$

$$A_t^{\text{low}} := \underbrace{\mathbb{E}[G_t \mid s_t, o_t, a_t]}_{Q^{\text{low}}(s_t, o_t, a_t)} - \underbrace{\mathbb{E}[G_t \mid s_t, o_t]}_{V^{\text{low}}(s_t, o_t)}.$$

*Proof.* First we define $Q$ functions and $V$ functions at different levels:

$$
\begin{aligned}
Q^{\text{high}}(s_t, o_t) &:= \mathbb{E}[G_t \mid s_t, q_t = 1, o_t], & V^{\text{high}}(s_t) &:= \mathbb{E}[G_t \mid s_t, q_t = 1], \\
Q^{\text{low}}(s_t, o_t, a_t) &:= \mathbb{E}[G_t \mid s_t, o_t, a_t], & V^{\text{low}}(s_t, o_t) &:= \mathbb{E}[G_t \mid s_t, o_t], \\
Q^{\text{switch}}(s_t, o_{t-1}, q_t) &:= \mathbb{E}[G_t \mid s_t, o_{t-1}, q_t], & V^{\text{switch}}(s_t, o_{t-1}) &:= \mathbb{E}[G_t \mid s_t, o_{t-1}].
\end{aligned}
$$
(17)

Recall the objective:

$$
\begin{aligned}
J(\theta) &= \mathbb{E}_{x \sim p(X)} \mathbb{E}_{\tau \sim \pi_\theta} \left[ \sum_{t=0}^{T-1} \gamma^t r_t \right] \\
&= \mathbb{E}_{x \sim p(X)} \left[ \sum_\tau p_\theta(\tau \mid x) \left( \sum_{t=0}^{T-1} \gamma^t r_t \right) \right].
\end{aligned}
$$

From the identity $\nabla_\theta p_\theta(x) = p_\theta(x) \nabla_\theta \log p_\theta(x)$, we have

$$\nabla_\theta J(\theta) = \mathbb{E}_{x \sim p(X)} \mathbb{E}_{\tau \sim \pi_\theta} \left[ \nabla_\theta \log p_\theta(\tau \mid x) \sum_{t=0}^{T-1} \gamma^t r_t \right].$$
(18)

Following the policy factorization in (2) and its realization by the LLM $\pi_\theta$, we have

$$p_\theta(\tau \mid x) = p(s_0 \mid x) \prod_{t=0}^{T-1} \left[ \pi_\theta(q_t \mid s_t, o_{t-1}) \big( (1 - q_t) \mathbf{1}[o_t = o_{t-1}] + q_t \pi_\theta(o_t \mid s_t) \big) \pi_\theta(a_t \mid s_t, o_t) p(s_{t+1}, r_t \mid s_t, a_t) \right]$$

$$= p(s_0 \mid x) \prod_{t=0}^{T-1} \left[ \pi_\theta(q_t \mid s_t, o_{t-1}) \, \pi_\theta(o_t \mid s_t)^{q_t} \, \pi_\theta(a_t \mid s_t, o_t) \, p(s_{t+1}, r_t \mid s_t, a_t) \right],$$
(19)

where $\mathbf{1}[\cdot]$ is the indicator function, and $p(s_{t+1}, r_t \mid s_t, a_t)$ is the environment transition.

Let $R_\tau := \sum_{t=0}^{T-1} \gamma^t r_t$, plug (19) into (18),

$$
\begin{aligned}
\nabla_\theta J(\theta) &= \mathbb{E}_x \mathbb{E}_\tau \left[ \sum_{t=0}^{T-1} \Big( \nabla_\theta \log \pi_\theta(q_t \mid s_t, o_{t-1}) + q_t \nabla_\theta \log \pi_\theta(o_t \mid s_t) + \nabla_\theta \log \pi_\theta(a_t \mid s_t, o_t) \Big) R_\tau \right] \\
&= \mathbb{E}_x \mathbb{E}_\tau \left[ \sum_{t=0}^{T-1} \Big( \nabla_\theta \log \pi_\theta(q_t \mid s_t, o_{t-1}) + q_t \nabla_\theta \log \pi_\theta(o_t \mid s_t) + \nabla_\theta \log \pi_\theta(a_t \mid s_t, o_t) \Big) G_t \right] \\
&= \mathbb{E}_x \mathbb{E}_\tau \left[ \sum_{t=0}^{T-1} \nabla_\theta \log \pi_\theta(q_t \mid s_t, o_{t-1}) G_t \right] + \mathbb{E}_x \mathbb{E}_\tau \left[ \sum_{t=0}^{T-1} q_t \nabla_\theta \log \pi_\theta(o_t \mid s_t) G_t \right] \\
&\quad + \mathbb{E}_x \mathbb{E}_\tau \left[ \sum_{t=0}^{T-1} \nabla_\theta \log \pi_\theta(a_t \mid s_t, o_t) G_t \right],
\end{aligned}
\tag{20}
$$

we can replace $R_\tau$ by $G_t$ because the prefix return $\sum_{k=0}^{t-1} \gamma^k r_k$ is independent of decisions at time $t$, and from the score function identity: $\mathbb{E}_{z \sim \pi_\theta(\cdot|c)}[\nabla_\theta \log \pi_\theta(z \mid c)\, b(c)] = 0$ for any $b(c)$ independent of $z$, so the prefix term vanishes in expectation.

Consider each term in (20) separately, first,

$$
\begin{aligned}
\mathbb{E}\left[ \sum_{t=0}^{T-1} \nabla_\theta \log \pi_\theta(q_t \mid s_t, o_{t-1}) \, G_t \right] &= \mathbb{E}\left[ \sum_{t=0}^{T-1} \nabla_\theta \log \pi_\theta(q_t \mid s_t, o_{t-1}) \mathbb{E}[G_t \mid s_t, o_{t-1}, q_t] \right] \\
&= \mathbb{E}\left[ \sum_{t=0}^{T-1} \nabla_\theta \log \pi_\theta(q_t \mid s_t, o_{t-1})\, Q^{\mathrm{switch}}(s_t, o_{t-1}, q_t) \right] \qquad \text{(by definition of } Q^{\mathrm{switch}}\text{)}.
\end{aligned}
$$

Similarly,

$$
\mathbb{E}\left[ \sum_{t=0}^{T-1} q_t \nabla_\theta \log \pi_\theta(o_t \mid s_t) \, G_t \right] = \mathbb{E}\left[ \sum_{t=0}^{T-1} q_t \nabla_\theta \log \pi_\theta(o_t \mid s_t) \mathbb{E}[G_t \mid s_t, q_t = 1, o_t] \right] = \mathbb{E}\left[ \sum_{t=0}^{T-1} q_t \nabla_\theta \log \pi_\theta(o_t \mid s_t)\, Q^{\mathrm{high}}(s_t, o_t) \right],
$$

and,

$$
\mathbb{E}\left[ \sum_{t=0}^{T-1} \nabla_\theta \log \pi_\theta(a_t \mid s_t, o_t) \, G_t \right] = \mathbb{E}\left[ \sum_{t=0}^{T-1} \nabla_\theta \log \pi_\theta(a_t \mid s_t, o_t) \mathbb{E}[G_t \mid s_t, o_t, a_t] \right] = \mathbb{E}\left[ \sum_{t=0}^{T-1} \nabla_\theta \log \pi_\theta(a_t \mid s_t, o_t)\, Q^{\mathrm{low}}(s_t, o_t, a_t) \right].
$$

Using the score function identity again: $\mathbb{E}_{z \sim \pi_\theta(\cdot|c)}[\nabla_\theta \log \pi_\theta(z \mid c)\, b(c)] = 0$ for any $b(c)$ independent of $z$, we can subtract:

$$
V^{\mathrm{switch}}(s_t, o_{t-1}) = \mathbb{E}[G_t \mid s_t, o_{t-1}], \quad V^{\mathrm{high}}(s_t) = \mathbb{E}[G_t \mid s_t, q_t = 1], \quad V^{\mathrm{low}}(s_t, o_t) = \mathbb{E}[G_t \mid s_t, o_t],
$$

from the corresponding $Q$ terms without changing the expectation. This yields

$$
\nabla_\theta J(\theta) = \mathbb{E}_{x \sim p(X)} \mathbb{E}_{\tau \sim \pi_\theta} \left[ \sum_{t=0}^{T-1} \Big( \nabla_\theta \log \pi_\theta(q_t \mid s_t, o_{t-1}) A_t^{\mathrm{switch}} + q_t \nabla_\theta \log \pi_\theta(o_t \mid s_t) A_t^{\mathrm{high}} + \nabla_\theta \log \pi_\theta(a_t \mid s_t, o_t) A_t^{\mathrm{low}} \Big) \right].
$$

The proof is completed. $\qquad \square$

### A.2. Proof of Theorem 4.2

Theorem 4.2 states that the gradient estimator obtained from the HAE process in Section 4.3 is an unbiased estimator of (5) up GAE bootstrapping and critic approximation errors. To see this, we consider two different gradient estimators and compare them with the true gradient $\nabla_\theta J(\theta)$:

First, $\hat{g}(\theta)$, which is the HAE gradient estimator, obtained by plugging $\{\hat{A}_t^{\mathrm{switch}}, \hat{A}_{b_k}^{\mathrm{high}}, \hat{A}_t^{\mathrm{low}}\}$ from (6), (8) and (10), with *learned* critics $V_\phi^{\mathrm{low}}$ and $V_\phi^{\mathrm{high}}$:

$$
\hat{g}(\theta) = \left[ \sum_{t=0}^{T-1} \Big( \nabla_\theta \log \pi_\theta(q_t \mid s_t, o_{t-1}) \, \hat{A}_t^{\mathrm{switch}} + q_t \nabla_\theta \log \pi_\theta(o_t \mid s_t) \hat{A}_t^{\mathrm{high}} + \nabla_\theta \log \pi_\theta(a_t \mid s_t, o_t) \hat{A}_t^{\mathrm{low}} \Big) \right], \tag{21}
$$

where $\{\hat{A}_t^{\text{switch}}, \hat{A}_{b_k}^{\text{high}}, \hat{A}_t^{\text{low}}\}$ are obtained from (6), (8) and (10), with learned critics $V_\phi^{\text{low}}$ and $V_\phi^{\text{high}}$;

Second, $g^\lambda(\theta)$, which is the corresponding oracle estimator of (5), obtained by using the same formulas in (6), (8), and (10) but with the *true* value functions $V^{\text{low}}, V^{\text{high}}$:

$$g^\lambda(\theta) = \left[ \sum_{t=0}^{T-1} \left( \nabla_\theta \log \pi_\theta(q_t \mid s_t, o_{t-1}) A_{\lambda,t}^{\text{switch}} + q_t \nabla_\theta \log \pi_\theta(o_t \mid s_t) \hat{A}_{\lambda,t}^{\text{high}} + \nabla_\theta \log \pi_\theta(a_t \mid s_t, o_t) \hat{A}_{\lambda,t}^{\text{low}} \right) \right], \quad (22)$$

where $\{A_{\lambda,t}^{\text{switch}}, A_{\lambda,b_k}^{\text{high}}, A_{\lambda,t}^{\text{low}}\}$ are obtained from (6), (8) and (10), with true value functions $V^{\text{low}}$ and $V^{\text{high}}$.

**Theorem A.2.** *The HAE gradient estimator $\hat{g}(\theta)$ is an unbiased gradient estimator up to GAE bootstrapping and critic approximation errors, that is,*

$$\mathbb{E}\big[\hat{g}(\theta)\big] - \nabla_\theta J(\theta) = \underbrace{\Big(\mathbb{E}\big[g^\lambda(\theta)\big] - \nabla_\theta J(\theta)\Big)}_{\text{GAE bootstrapping bias}} + \underbrace{\mathbb{E}\big[\hat{g}(\theta) - g^\lambda(\theta)\big]}_{\text{critic approximation bias}}. \quad (23)$$

*In particular, when $\lambda^{low} = \lambda^{high} = 1$ (Monte Carlo estimation, GAE bootstrapping bias =0) and $V_\phi^{low} = V^{low}, V_\phi^{high} = V^{high}$ (critics are perfectly learned, critic approximation bias=0), $\mathbb{E}[\hat{g}(\theta)] = \nabla_\theta J(\theta)$, i.e. $\hat{g}(\theta)$ is an unbiased stochastic gradient estimator of $\nabla_\theta J(\theta)$.*

*Proof.* The decomposition in (23) is straightforward by linearity of expectation:

$$\begin{aligned}
\mathbb{E}\big[\hat{g}(\theta)\big] - \nabla_\theta J(\theta) &= \mathbb{E}\big[\hat{g}(\theta)\big] - \nabla_\theta J(\theta) + \mathbb{E}[g^\lambda(\theta)] - \mathbb{E}[g^\lambda(\theta)] \\
&= \Big(\mathbb{E}\big[g^\lambda(\theta)\big] - \nabla_\theta J(\theta)\Big) + \mathbb{E}\big[\hat{g}(\theta) - g^\lambda(\theta)\big].
\end{aligned}$$

Next we show unbiasedness when $\lambda^{\text{low}} = \lambda^{\text{high}} = 1$ and critics are perfectly learned, by verifying that, under these conditions, each advantage estimator $\hat{A}_t^{\text{low}}, \hat{A}_t^{\text{high}}, \hat{A}_t^{\text{switch}}$ is unbiased.

Recall $0 = b_0 < b_1 < \cdots < b_K = T$ are the switching boundary indices, where $T$ is the episode length and $K$ is the number of segments, and let the $k$-th segment be the time interval $t \in [b_k, b_{k+1} - 1]$ during which subgoal $o_k$ is active.

**Low-level** ($\hat{A}_t^{\text{low}}$). For step $t$ within segment $k$, i.e. $b_k \leq t < b_{k+1}$, when $\lambda^{\text{low}} = 1$, and $V_\phi^{\text{high}} = V^{\text{high}}$ and $V_\phi^{\text{low}} = V^{\text{low}}$, from (6), we have

$$\hat{A}_t^{\text{low}} = \sum_{\ell=t}^{b_{k+1}-1} \gamma^\ell \delta_\ell^{\text{low}}, \quad (24)$$

Recall the low-level TD residual is defined in (6) as:

$$\delta_t^{\text{low}} = r_t + \gamma V_t^{\text{next}} - V^{\text{low}}(s_t, o_k), \; t \in [b_k, b_{k+1} - 1],$$

where

$$V_t^{\text{next}} = \begin{cases} V^{\text{high}}(s_{b_{k+1}}), & t = b_{k+1} - 1 \\ V^{\text{low}}(s_{t+1}, o_k), & \text{otherwise.} \end{cases}$$

Telescoping the sum in (24) and by definition of low-level TD residual, we obtain

$$\hat{A}_t^{\text{low}} = \sum_{\ell=t}^{b_{k+1}-1} \gamma^{\ell-t} r_\ell + \gamma^{b_{k+1}-t} V^{\text{high}}(s_{b_{k+1}}) - V^{\text{low}}(s_t, o_t).$$

Taking expectation, and by definition of $V^{\text{high}}$ and $V^{\text{low}}$ in (17),

$$
\begin{aligned}
\mathbb{E}[\hat{A}_t^{\text{low}} \mid s_t, o_t, a_t] &= \mathbb{E}\left[ \sum_{\ell=t}^{b_{k+1}-1} \gamma^{\ell-t} r_\ell \mid s_t, o_t, a_t \right] + \mathbb{E}[\gamma^{b_{k+1}-t}\mathbb{E}[G_{b_{k+1}} \mid s_{b_{k+1}}, q_{b_{k+1}} = 1] \mid s_t, o_t, a_t] - \mathbb{E}[G_t \mid s_t, o_t] \\
&= \mathbb{E}\left[ \sum_{\ell=t}^{b_{k+1}-1} \gamma^{\ell-t} r_\ell \; + \; \gamma^{b_{k+1}-t}\mathbb{E}[G_{b_{k+1}} \mid s_{b_{k+1}}, q_{b_{k+1}} = 1] \mid s_t, o_t, a_t \right] - \mathbb{E}[G_t \mid s_t, o_t] \\
&= \mathbb{E}[G_t \mid s_t, o_t, a_t] - \mathbb{E}[G_t \mid s_t, o_t] \\
&= A_t^{\text{low}}
\end{aligned}
$$

where the third equality is by law of iterated expectations.

**High-level** ($\hat{A}_{b_k}^{\text{high}}$). For the $k$-th segment boundary time $b_k$ (i.e., $q_{b_k} = 1$), when $\hat{V}^{\text{high}} = V^{\text{high}}$, the high-level GAE in (8) reduces to a sum of high-level TD residuals along the boundary-indexed process:

$$
\hat{A}_{b_k}^{\text{high}} = \sum_{j=k}^{K-1} \left( \gamma^{b_j - b_k} \right) \delta_{b_j}^{\text{high}}, \tag{25}
$$

Recall the high-level TD residual is

$$
\delta_{b_j}^{\text{high}} := R_{b_j} + \gamma^{b_{j+1}-b_j} V^{\text{high}}(s_{b_{j+1}}) - V^{\text{high}}(s_{b_j}), \qquad R_{b_j} := \sum_{u=b_j}^{b_{j+1}-1} \gamma^{u-b_j} r_u.
$$

Telescoping the sum in (25) yields

$$
\hat{A}_{b_k}^{\text{high}} = \sum_{j=k}^{K-1} \gamma^{b_j - b_k} R_{b_j} - V^{\text{high}}(s_{b_k}) = \sum_{u=b_k}^{T-1} \gamma^{u-b_k} r_u - V^{\text{high}}(s_{b_k}) = G_{b_k} - V^{\text{high}}(s_{b_k}).
$$

Taking expectation and by definition of $V^{\text{high}}$ in (17),

$$
\begin{aligned}
\mathbb{E}[\hat{A}_{b_k}^{\text{high}} \mid s_{b_k}, q_{b_k} = 1, o_{b_k}] &= \mathbb{E}[G_{b_k} \mid s_{b_k}, q_{b_k} = 1, o_{b_k}] - \mathbb{E}[G_{b_k} \mid s_{b_k}, q_{b_k} = 1] \\
&= Q^{\text{high}}(s_{b_k}, o_{b_k}) - V^{\text{high}}(s_{b_k}) = A_{b_k}^{\text{high}}.
\end{aligned}
$$

**Switching** ($\hat{A}_t^{\text{switch}}$). Recall the switching gain is

$$
\delta_t^{\text{switch}} := V^{\text{high}}(s_t) - V^{\text{low}}(s_t, o_{t-1}), \qquad \beta_t := \pi_\theta(q_t = 1 \mid s_t, o_{t-1}),
$$

and the estimator in (10) is

$$
\hat{A}_t^{\text{switch}} = (q_t - \beta_t)\, \delta_t^{\text{switch}}.
$$

Under the Plan-Execute semantics at the same state $s_t$: continuing ($q_t = 0$) keeps option $o_{t-1}$, while terminating ($q_t = 1$) returns to the high-level process, hence

$$
Q^{\text{switch}}(s_t, o_{t-1}, 0) = V^{\text{low}}(s_t, o_{t-1}), \qquad Q^{\text{switch}}(s_t, o_{t-1}, 1) = V^{\text{high}}(s_t).
$$

Therefore

$$
V^{\text{switch}}(s_t, o_{t-1}) = (1 - \beta_t) V^{\text{low}}(s_t, o_{t-1}) + \beta_t V^{\text{high}}(s_t).
$$

Now consider the true switching advantage:

$$
\begin{aligned}
A_t^{\text{switch}} &:= Q^{\text{switch}}(s_t, o_{t-1}, q_t) - V^{\text{switch}}(s_t, o_{t-1}) \\
&= \left( (1 - q_t) V^{\text{low}}(s_t, o_{t-1}) + q_t V^{\text{high}}(s_t) \right) - \left( (1 - \beta_t) V^{\text{low}}(s_t, o_{t-1}) + \beta_t V^{\text{high}}(s_t) \right) \\
&= (q_t - \beta_t) \left( V^{\text{high}}(s_t) - V^{\text{low}}(s_t, o_{t-1}) \right) \\
&= (q_t - \beta_t) \delta_t^{\text{switch}} = \hat{A}_t^{\text{switch}}.
\end{aligned}
$$

Hence, under perfect critics, $\hat{A}_t^{\text{switch}}$ equals the true switching advantage.

Combining the above three unbiased advantages and summing over $t$ yields

$$\mathbb{E}[\hat{g}(\theta)] = \mathbb{E}\left[\sum_{t=0}^{T-1}\left(\nabla_\theta \log \pi_\theta(q_t \mid s_t, o_{t-1})\, A_t^{\text{switch}} + q_t\, \nabla_\theta \log \pi_\theta(o_t \mid s_t)\, A_t^{\text{high}} + \nabla_\theta \log \pi_\theta(a_t \mid s_t, o_t)\, A_t^{\text{low}}\right)\right] = \nabla_\theta J(\theta).$$

The proof is completed. $\qquad\square$

### A.3. Formal Theorem and Proof of Theorem 4.3

**Theorem A.3.** *Under a fixed Plan-Execute policy $\pi_\theta$ and its on-policy rollout distribution. Let $G_t := \sum_{t'=t}^{T-1} \gamma^{t'-t} r_{t'}$ be the return-to-go. For step $t$ within segment $k$, i.e. $b_k \le t < b_{k+1}$, during which subgoal $o_k$ is active, consider two advantage estimation schemes below:*

- **Advantage from flat GAE.** $A_t^{flat} := \sum_{\ell=t}^{T-1}(\gamma\lambda^{flat})^{\ell-t}\delta_\ell^{flat}$, where $\delta_\ell^{flat} := r_\ell + \gamma \hat{V}^{flat}(s_{\ell+1}) - \hat{V}^{flat}(s_\ell)$, and $V^{flat}$ is a state-only critic. In particular, when $\lambda^{flat} = 1$ and $\hat{V}^{flat}(s_t) = V^{flat}(s_t) := \mathbb{E}[G_t \mid s_t]$, $A_t^{flat} = G_t - V^{flat}(s_t)$.

- **Low-level Advantage from HAE.** $A_t^{low}$, as defined in (6), with critics $\hat{V}^{high}$ and $\hat{V}^{low}$. In particular, when $\lambda^{low} = 1$ and $\hat{V}^{high}(s_t) = V^{high}(s_t)$, $\hat{V}^{low}(s_t, o_t) = V^{low}(s_t, o_t)$, $A_t^{low} = \bar{G}_t - V^{low}(s_t, o_t)$, where $\bar{G}_t := \sum_{\ell=t}^{b_{k+1}-1} \gamma^{\ell-t} r_\ell + \gamma^{b_{k+1}-t} V^{high}(s_{b_{k+1}})$.

*Note that both advantage estimation schemes operate under the Plan-Execute policy.*

**Claim.** *Assume $\mathbb{E}[G_t^2] < \infty$, when $\lambda^{flat} = \lambda^{high} = \lambda^{low} = 1$, and values baselines are exact, i.e., $\hat{V}^{flat}(s_t) = V^{flat}(s_t)$, $\hat{V}^{low}(s_t, o_t) = V^{low}(s_t, o_t)$, $\hat{V}^{high}(s_t) = V^{high}(s_t)$, then for any fixed time index $t$,*

$$Var\left(A_t^{low}\right) \le Var\left(A_t^{flat}\right).$$

*Proof.* We show that HAE reduces variance through two channels: boundary bootstrapping and option-conditioned baseline.

**Boundary bootstrapping.** Define

$$\mathcal{Z} := \sigma(s_t, o_t), \qquad \mathcal{W} := \sigma\left(\mathcal{Z}, r_t, \ldots, r_{b_{k+1}-1}, b_{k+1}, s_{b_{k+1}}, q_{b_{k+1}}\right),$$

so $\mathcal{Z} \subseteq \mathcal{W}$. We have

$$\bar{G}_t := R_{t:b_{k+1}-1} + \gamma^{b_{k+1}-t} V^{high}(s_{b_{k+1}}) = R_{t:b_{k+1}-1} + \gamma^{b_{k+1}-t}\mathbb{E}[G_{b_{k+1}} \mid s_{b_{k+1}}, q_{b_{k+1}} = 1] = \mathbb{E}[G_t \mid \mathcal{W}],$$

where $R_{t:b_{k+1}-1} := \sum_{\ell=t}^{b_{k+1}-1} \gamma^{\ell-t} r_\ell$. By law of total expectation,

$$\mathbb{E}[\bar{G}_t \mid \mathcal{Z}] = \mathbb{E}[\mathbb{E}[G_t \mid \mathcal{W}] \mid \mathcal{Z}] = \mathbb{E}[G_t \mid \mathcal{Z}].$$

Meanwhile,

$$A_t^{low} = \bar{G}_t - V^{low}(s_t, o_t) = \mathbb{E}[G_t \mid \mathcal{W}] - \mathbb{E}[G_t \mid \mathcal{Z}]$$

Denote $Z := G_t - \mathbb{E}[G_t \mid \mathcal{Z}]$. Then $A_t^{low} = \mathbb{E}[Z \mid \mathcal{W}]$.

By law of total variance, we have $\text{Var}(Z) = \mathbb{E}[\text{Var}(Z \mid \mathcal{W})] + \text{Var}(\mathbb{E}[Z \mid \mathcal{W}])$, hence

$$\text{Var}(A_t^{low}) = \text{Var}\left(\mathbb{E}[Z \mid \mathcal{W}]\right) \le \text{Var}(Z) = \text{Var}\left(G_t - \mathbb{E}[G_t \mid s_t, o_t]\right). \qquad (26)$$

**Option-conditioned Baseline.** Let $\mathcal{F} := \sigma(s_t)$ and recall $\mathcal{Z} = \sigma(s_t, o_t)$. By the conditional-variance identity

$$\text{Var}\left(G_t - \mathbb{E}[G_t \mid \mathcal{H}]\right) = \mathbb{E}\left[\text{Var}(G_t \mid \mathcal{H})\right] \qquad \text{for any } \sigma\text{-field } \mathcal{H},$$

we have $\text{Var}\left(G_t - \mathbb{E}[G_t \mid s_t, o_t]\right) = \mathbb{E}[\text{Var}(G_t \mid \mathcal{Z})]$. By law of total variance,

$$\text{Var}(G_t \mid \mathcal{F}) = \mathbb{E}\left[\text{Var}(G_t \mid \mathcal{Z}) \mid \mathcal{F}\right] + \text{Var}\left(\mathbb{E}[G_t \mid \mathcal{Z}] \mid \mathcal{F}\right).$$

Taking expectation over $\mathcal{F}$,

$$\mathbb{E}\big[\mathrm{Var}(G_t \mid \mathcal{F})\big] = \mathbb{E}\big[\mathrm{Var}(G_t \mid \mathcal{Z})\big] + \mathbb{E}\big[\mathrm{Var}\big(\mathbb{E}[G_t \mid \mathcal{Z}] \mid \mathcal{F}\big)\big] \geq \mathbb{E}\big[\mathrm{Var}(G_t \mid \mathcal{Z})\big]$$

Using the conditional-variance identity again,

$$\mathrm{Var}\big(G_t - \mathbb{E}[G_t \mid s_t, o_t]\big) \leq \mathrm{Var}\big(G_t - \mathbb{E}[G_t \mid s_t]\big). \tag{27}$$

Note that $A_t^{\mathrm{flat}} = G_t - \mathbb{E}[G_t \mid s_t]$, combining (26) and (27), it follows that $\mathrm{Var}\big(A_t^{\mathrm{low}}\big) \leq \mathrm{Var}\big(A_t^{\mathrm{flat}}\big)$.

$\square$

# B. Algorithm Details

---

**Algorithm 1** Training LLM Agents with HiPER

---

**Require:** Initial policy $\pi_{\theta_{\mathrm{old}}}$; two-head critic $V_\phi = \{V_\phi^{\mathrm{high}}, V_\phi^{\mathrm{low}}\}$; task distribution $p(X)$; discount $\gamma$; GAE parameters $\lambda^{\mathrm{high}}, \lambda^{\mathrm{low}}$; clipping $\varepsilon$; KL penalty $\beta$.

1: **for** each training iteration **do**
2:     Update old policy: $\theta_{\mathrm{old}} \leftarrow \theta$
3:     // Plan-Execute rollout
4:     Sample task $x \sim p(X)$ and collect trajectory $\tau = \{(s_t, o_{t-1}, q_t, o_t, a_t, r_t, s_{t+1})\}_{t=0}^{T-1}$ by
5:     **for** $t = 0$ to $T - 1$ **do**
6:         Sample $q_t \sim \pi_{\theta_{\mathrm{old}}}(\cdot \mid s_t, o_{t-1})$
7:         **if** $q_t = 1$ **then**
8:             Sample $o_t \sim \pi_{\theta_{\mathrm{old}}}(\cdot \mid s_t)$
9:         **else**
10:            $o_t \leftarrow o_{t-1}$
11:         **end if**
12:         Sample $a_t \sim \pi_{\theta_{\mathrm{old}}}(\cdot \mid s_t, o_t)$
13:         Execute $a_t$, observe $r_t, s_{t+1}$
14:     **end for**
15:     // Hierarchical Advantage Estimation and critic targets
16:     Identify segment boundaries $\{b_k\}_{k=0}^K$ from switches $\{t : q_t = 1\}$ (with $b_0 = 0, b_K = T$)
17:     Estimate hierarchical advantages $\{\hat{A}_t^{\mathrm{high}}, \hat{A}_t^{\mathrm{switch}}, \hat{A}_t^{\mathrm{low}}\}$ via Equations (6)–(10)
18:     Compute critic targets $\{y_k^{\mathrm{high}}\}_{k=0}^{K-1}$ and $\{y_t^{\mathrm{low}}\}_{t=0}^{T-1}$ via Equations. (11)–(13)
19:     // PPO-style update
20:     Form PPO ratios $r_t^{\mathrm{switch}}(\theta), r_t^{\mathrm{high}}(\theta), r_t^{\mathrm{low}}(\theta)$, clipped surrogates $\mathcal{L}_{\mathrm{clip}}^{\mathrm{switch}}, \mathcal{L}_{\mathrm{clip}}^{\mathrm{high}}, \mathcal{L}_{\mathrm{clip}}^{\mathrm{low}}$
21:     $\mathcal{L}_{\mathrm{actor}}(\theta) \leftarrow \mathcal{L}_{\mathrm{clip}}^{\mathrm{low}}(\theta) + \mathcal{L}_{\mathrm{clip}}^{\mathrm{switch}}(\theta) + \mathcal{L}_{\mathrm{clip}}^{\mathrm{high}}(\theta)$
22:     $\mathcal{L}_V(\phi) \leftarrow \sum_{t=0}^{T-1}\big(V_\phi^{\mathrm{low}}(s_t, o_t) - y_t^{\mathrm{low}}\big)^2 + \sum_{k=0}^{K-1}\big(V_\phi^{\mathrm{high}}(s_{b_k}) - y_k^{\mathrm{high}}\big)^2$
23:     Update $(\theta, \phi)$ by minimizing the PPO loss in Eq. (30):

$$\min_{\theta,\phi} \; -\mathcal{L}_{\mathrm{actor}}(\theta) + c_V \mathcal{L}_V(\phi) - \beta\, \mathbb{D}_{\mathrm{KL}}\big(\pi_\theta(\cdot \mid x) \,\|\, \pi_{\mathrm{ref}}(\cdot \mid x)\big).$$

24: **end for**

---

**PPO Loss.** Recall the Plan-Execute policy in (2),

$$\pi_\theta(q_t, o_t, a_t \mid s_t, o_{t-1}) = \pi_\theta^{\mathrm{switch}}(q_t \mid s_t, o_{t-1}) \cdot \Big(\pi_\theta^{\mathrm{high}}(o_t \mid s_t)\Big)^{q_t} \cdot \pi_\theta^{\mathrm{low}}(a_t \mid s_t, o_t).$$

We define the following probability ratios used for PPO clipping,

$$r_t^{\mathrm{switch}}(\theta) = \frac{\pi_\theta^{\mathrm{switch}}(q_t \mid s_t, o_{t-1})}{\pi_{\theta_{\mathrm{old}}}^{\mathrm{switch}}(q_t \mid s_t, o_{t-1})}, \qquad r_t^{\mathrm{high}}(\theta) = \frac{\pi_\theta^{\mathrm{high}}(o_t \mid s_t)}{\pi_{\theta_{\mathrm{old}}}^{\mathrm{high}}(o_t \mid s_t)}, \qquad r_t^{\mathrm{low}}(\theta) = \frac{\pi_\theta^{\mathrm{low}}(a_t \mid s_t, o_t)}{\pi_{\theta_{\mathrm{old}}}^{\mathrm{low}}(a_t \mid s_t, o_t)},$$

and the clipped surrogate for each decision (switching, subgoal, action):

$$\mathcal{L}_{\text{clip}}^{z}(\theta) = \mathbb{E}\left[\sum_{t=0}^{T-1} \min\left(r_t^z(\theta)\,\hat{A}_t^z,\; \text{clip}_{\varepsilon}(r_t^z(\theta))\,\hat{A}_t^z\right)\right], \qquad z \in \{\text{switch}, \text{low}\},$$

$$\mathcal{L}_{\text{clip}}^{\text{high}}(\theta) = \mathbb{E}\left[\sum_{t=0}^{T-1} q_t \min\left(r_t^{\text{high}}(\theta)\,\hat{A}_t^{\text{high}},\; \text{clip}_{\varepsilon}(r_t^{\text{high}}(\theta))\,\hat{A}_t^{\text{high}}\right)\right].$$

The combined actor loss is therefore

$$\mathcal{L}_{\text{actor}}(\theta) = \mathcal{L}_{\text{clip}}^{\text{low}}(\theta) + \mathcal{L}_{\text{clip}}^{\text{switch}}(\theta) + \mathcal{L}_{\text{clip}}^{\text{high}}(\theta). \tag{28}$$

Assume the two-head critic model is parameterized by $\phi$, given the critic targets $y_k^{\text{high}}$ and $y_t^{\text{low}}$ defined in (11) and (13),

$$\mathcal{L}_V(\phi) = c_V\left(\mathbb{E}\left[\sum_{t=0}^{T-1}\left(V_\phi^{\text{low}}(s_t, o_k) - y_t^{\text{low}}\right)^2\right] + \mathbb{E}\left[\sum_{k=0}^{K-1}\left(V_\phi^{\text{high}}(s_{b_k}) - y_k^{\text{high}}\right)^2\right]\right), \tag{29}$$

Combining the actor and critic losses, the overall PPO objective is

$$\min_{\theta,\phi}\ \mathcal{L}(\theta,\phi) = \mathbb{E}_{x\sim p(X),\tau\sim\pi_\theta}\left[-\mathcal{L}_{\text{actor}}(\theta) + c_V\mathcal{L}_V(\phi) - \beta\mathbb{D}_{\text{KL}}(\pi_\theta(\cdot\mid x)\|\pi_{\text{ref}}(\cdot\mid x))\right], \tag{30}$$

where $\beta$ controls the KL penalty strength to encourage proximity to a reference model.

**Reward Design.** In addition to the reward provided in the original environments (e.g. outcome reward $R = 10$ for success in ALFWorld and WebShop), we add a format penalty to penalize invalid format generated by the agent. Concretely, if the agent fails to follow the {<think>...</think><action>...</action>} (for ReAct prompting) or the {<switch>...</switch><subgoal>...</subgoal><action>...</action>} (for Plan-Execute prompting) format at turn index $t$, we incur a 0.1 penalty to $r_t$. Moreover, specific to HiPER, we incur a small penalty $c_t = c_{\text{KEEP}}(1 - q_t)$ to each turn where the agent decides to KEEP, so as to encourage exploration at the high-level and prevent degenerate behavior, such as committing to a single subgoal for an entire episode. Empirically we find this helpful to boost performance, and results are fairly robust to the choice of $c_{\text{KEEP}}$, as shown in Appendix D.3.

## C. Implementation Details

### C.1. Hyperparameters

**ALFWorld.** All methods use the same hyperparameter settings: a maximum prompt length of 2048 tokens, a maximum response length of 512 tokens, and up to 50 environment steps per episode. Learning rate for the actor is set to 1e-6, and learning rate for the critic is set to 1e-5 (used for PPO and HiPER). For group-based RL (GRPO, RLOO. GiGPO), the group size is set to 8 and 16 distinct groups are sampled per rollout, resulting in 128 environments in total. For HiPER and PPO, we collect rollouts from 128 independent environments. The rollout temperature is 1.0, and the validation temperature is 0.4. We use a mini-batch size of 256 and set the KL-divergence loss coefficient to 0.01. For HiPER, both $\lambda^{\text{high}}$ and $\lambda^{\text{low}}$ are set to 0.95 without additional tuning.

**WebShop.** All methods use the same hyperparameter settings: a maximum prompt length of 4096 tokens, a maximum response length of 512 tokens, and up to 15 environment steps per episode. Learning rate for the actor is set to 1e-6, and learning rate for the critic is set to 1e-5 (used for PPO and HiPER). For group-based RL (GRPO, RLOO. GiGPO), the group size is set to 8 and 16 distinct groups are sampled per rollout, resulting in 128 environments in total. For HiPER and PPO, we collect rollouts from 128 independent environments. The rollout temperature is 1.0, and the validation temperature is 0.4. We use a mini-batch size of 256 and set the KL-divergence loss coefficient to 0.01. For HiPER, both $\lambda^{\text{high}}$ and $\lambda^{\text{low}}$ are set to 0.95 without additional tuning.

## C.2. Memory Overhead

HiPER incurs negligible GPU memory overhead relative to standard PPO. Although HiPER uses two value estimates (high- and low-level), we implement them via a single shared critic backbone with two output heads, rather than training two separate critics. The extra parameters (and optimizer states) introduced by the additional head are tiny compared to the full model, so overall GPU memory is largely dominated by components shared with PPO, such as the actor/critic backbones, activations, and rollout buffers. In practice, small additional memory can also arise from non-model factors, e.g., longer effective sequence lengths due to structured Plan–Execute outputs (switch/subgoal/action), increased token-level bookkeeping, or rollout-generation KV caching. Empirically, under identical training settings and measured by peak GPU memory allocated, HiPER uses around $0.8\%$ more memory per GPU than flat PPO.

## C.3. Prompt Templates

In Table 1, we use the ReAct (Yao et al., 2022b) prompt strategy for baselines (Base Model, PPO, RLOO, GRPO, GiGPO), and use Plan-Execute prompt strategy for HiPER.

**ReAct vs Plan-Execute.** In ReAct prompting, the agent is instructed to reason about the current situation and take an action at each step, effectively operating on a single timescale. After each observation, the agent reasons again from scratch to determine the next action. From a modeling perspective, this stepwise reason-action interleaving makes the agent's global intent largely implicit and inconsistent, which could potentially lead to undesired behaviors such as intent drifting and repetitive actions. From a learning perspective, under the single-timescale structure, the agent must infer implicit long-range dependencies from sparse and delayed feedback, and assign credit along long-horizon action trajectories. Plan–Execute, by contrast, introduces an explicit temporal abstraction: the agent first commits to a high-level plan or subgoal that is meant to persist for multiple steps, and then conditions its low-level actions on that stable subgoal until a deliberate switch/replan boundary is triggered. This separation allows for more effective modeling and more efficient learning for multi-turn agents. For modeling, it externalizes the current global intent (subgoal) that stabilizes behavior, reduces goal drift across many steps rather than reasoning about the situation over again each turn. For learning, the separation of high-level planning and low-level execution allows for explicit and joint optimization of both planning and execution. In the meantime, this separation provides the structural basis for hierarchical advantage estimation by defining well-formed segments and boundary conditions, allowing learning signals for planning, execution and switching to be computed on their appropriate timescales and coupled in a principled way, enabling more targeted and efficient credit assignment.

---

**ReAct Prompt Template for ALFWorld**

You are an expert agent operating in the ALFRED embodied Environment. Your task is to: {task_description}. Prior to this step, you have already taken {step_count} step(s). Below are the most recent {history_length} observations and the corresponding actions you took: {action_history}. You are now at step {current_step} and your current observation is: {current_observation}. Your admissible actions of the current situation are: [{admissible_actions}]. Now it's your turn to take an action. You should first reason step-by-step about the current situation. This reasoning process MUST be enclosed within <think> </think> tags. Once you've finished your reasoning, you should choose an admissible action for current step and present it within <action> </action> tags.

*Figure 4.* ReAct prompt template of ALFWorld agents.

---

**Plan-Execute Prompt Template for ALFWorld**

You are an expert agent operating in the ALFRED Embodied Environment. Your overall task is: {task_description}
You will complete the task by maintaining a SHORT-TERM subgoal at each step. A subgoal is a small high-level objective that can typically be achieved in a few actions. A subgoal is NOT the full task and NOT a low-level action. At every step, you reconsider your current subgoal based on the latest observation, and may continue it or switch to a new short-term subgoal.

You have already taken {step_count} step(s). Below are the most recent {history_length} observations and actions you took: {action_history}. You are now at step {current_step} and your current observation is: {current_observation}. Your current subgoal is: {current_subgoal}. Your admissible actions are: [{admissible_actions}].

You MUST output EXACTLY THREE blocks, in the order shown below:
1) A <switch>... </switch> block (KEEP or SWITCH); 2) A <subgoal>... </subgoal> block (the subgoal to follow next); 3) A <action>... </action> block (one admissible action).

Format requirements:
- <switch> MUST contain only "KEEP" or "SWITCH".
- <subgoal> MUST appear at every step: If you KEEP, copy the exact current subgoal into <subgoal>. If you SWITCH, write a new short subgoal achievable in a few actions and not the entire task.
- <action> MUST contain exactly action verbatim copied as is from the admissible actions list.

*Figure 5.* Plan-Execute prompt template of ALFWorld agents.

**ReAct Prompt Template for WebShop**

You are an expert autonomous agent operating in the WebShop e-commerce environment. Your task is to: {task_description}. Prior to this step, you have already taken {step_count} step(s). Below are the most recent {history_length} observations and the corresponding actions you took: {action_history}. You are now at step {current_step} and your current observation is: {current_observation}. Your admissible actions of the current situation are: [{admissible_actions}].

Now it's your turn to take an action. You should first reason step-by-step about the current situation. This reasoning process MUST be enclosed within <think> </think> tags. Once you've finished your reasoning, you should choose an admissible action for current step and present it within <action> </action> tags.

*Figure 6.* ReAct prompt template of WebShop agents.

---

**Plan-Execute Prompt Template for WebShop**

You are an expert autonomous agent operating in the WebShop e-commerce environment. Your overall task is: {task_description}

You will complete the task by maintaining a SHORT-TERM subgoal at each step. A subgoal is a small high-level objective that can typically be achieved in a few actions. A subgoal is NOT the full task and NOT a low-level action. At every step, you reconsider your current subgoal based on the latest observation, and may continue it or switch to a new short-term subgoal.

You have already taken {step_count} step(s). Below are the most recent {history_length} observations and actions you took: {action_history}. You are now at step {current_step} and your current observation is: {current_observation}. Your current subgoal is: {current_subgoal}. Your admissible actions are: [{admissible_actions}].

You MUST output EXACTLY THREE blocks, in the order shown below:

1) A <switch>... </switch> block (KEEP or SWITCH); 2) A <subgoal>... </subgoal> block (the subgoal to follow next); 3) A <action>... </action> block (one admissible action).

Format requirements:

- <switch> MUST contain only "KEEP" or "SWITCH".

- <subgoal> MUST appear at every step: If you KEEP, copy the exact current subgoal into <subgoal>. If you SWITCH, write a new short subgoal achievable in a few actions and not the entire task.

- <action> MUST contain exactly action verbatim copied as is from the admissible actions list.

*Figure 7.* Plan-Execute prompt template of WebShop agents.

## D. Additional Results

### D.1. Ablation on Plan-Execute

*Table 2.* **Comparison of ReAct and Plan-Execute prompting on ALFWorld.** The upper panel reports performance of baselines when trained and evaluated with ReAct prompting; the lower panel reports performance of baselines and HiPER when trained and evaluated with Plan-Execute prompting. The base model used here is Qwen2.5-1.5B-Instruct. We report the success rate (%) of all six task categories, and the overall success rate. Results are averaged over 3 random seeds. The best values are in **bold**, and second best values are underlined.

| Method | ALFWorld | | | | | | |
| --- | --- | --- | --- | --- | --- | --- | --- |
| | Pick | Look | Clean | Heat | Cool | Pick2 | All |
| Base Model (ReAct) | $15.9_{\pm1.8}$ | $13.9_{\pm6.7}$ | $11.2_{\pm4.1}$ | $3.5_{\pm6.1}$ | $0.0_{\pm0.0}$ | $4.2_{\pm0.7}$ | $8.3_{\pm0.9}$ |
| +PPO | $74.0_{\pm9.0}$ | $37.5_{\pm21.7}$ | $67.0_{\pm5.1}$ | $85.6_{\pm17.1}$ | $68.8_{\pm3.7}$ | $56.1_{\pm12.2}$ | $68.2_{\pm1.8}$ |
| +GRPO | $77.4_{\pm6.7}$ | $54.2_{\pm7.2}$ | $75.6_{\pm8.6}$ | $85.6_{\pm17.1}$ | $67.8_{\pm7.4}$ | $56.1_{\pm11.0}$ | $71.1_{\pm8.2}$ |
| +GiGPO[†] | $94.4_{\pm5.9}$ | $67.5_{\pm4.6}$ | $94.8_{\pm3.8}$ | $\mathbf{94.4}_{\pm7.8}$ | $\underline{79.8}_{\pm4.7}$ | $76.4_{\pm5.4}$ | $86.7_{\pm1.7}$ |
| Base Model (Plan-Execute) | $3.4_{\pm3.3}$ | $10.4_{\pm3.6}$ | $3.0_{\pm2.6}$ | $0.0_{\pm0.0}$ | $0.0_{\pm0.0}$ | $1.3_{\pm2.3}$ | $2.9_{\pm1.6}$ |
| +PPO | $89.6_{\pm6.9}$ | $73.5_{\pm4.4}$ | $86.9_{\pm13.5}$ | $85.3_{\pm6.6}$ | $72.2_{\pm6.9}$ | $68.6_{\pm3.8}$ | $81.3_{\pm4.7}$ |
| +GRPO | $83.5_{\pm4.0}$ | $54.2_{\pm7.2}$ | $60.8_{\pm13.8}$ | $80.0_{\pm10.0}$ | $62.9_{\pm9.7}$ | $51.6_{\pm12.8}$ | $69.8_{\pm6.4}$ |
| +GiGPO | $\mathbf{99.1}_{\pm1.5}$ | $\underline{83.3}_{\pm14.9}$ | $\mathbf{98.8}_{\pm2.1}$ | $81.0_{\pm4.1}$ | $76.8_{\pm11.8}$ | $\mathbf{92.9}_{\pm0.1}$ | $\underline{91.1}_{\pm3.6}$ |
| **+HiPER** | $\underline{98.9}_{\pm1.9}$ | $\mathbf{91.7}_{\pm14.4}$ | $\underline{97.5}_{\pm4.3}$ | $\underline{90.9}_{\pm4.7}$ | $\mathbf{96.7}_{\pm2.8}$ | $\underline{91.3}_{\pm9.6}$ | $\mathbf{95.3}_{\pm1.4}$ |

### D.2. Critic Model Size

HiPER, like PPO, relies on a learned value function to form low-variance advantage estimates, and therefore requires a separate critic model. This introduces additional GPU memory consumption compared to critic-free baselines such as GRPO and GiGPO. In practice, however, the critic's prediction task (estimating scalar value targets) is substantially simpler than the actor's generative task, which suggests that a smaller critic may be sufficient while reducing memory overhead. Here, we test the performance of HiPER with different critic backbone sizes while keeping the actor model size fixed.

The results in Table 3 suggest that HiPER does not require a critic model that strictly matches the actor in size. Using a moderately smaller critic model for HiPER, the performance remains competitive. This could be a potential practical way to

*Table 3.* **Comparison of different critic model sizes on ALFWorld.** Rows denote actor-critic size pairs (e.g., HiPER-7B-1.5B uses a 7B actor with a 1.5B critic). We report the success rate (%) of all six task categories, and the overall success rate. Results are averaged over 3 random seeds. The best values are in **bold**, and second best values are underlined.

| Method | ALFWorld | | | | | | |
| --- | --- | --- | --- | --- | --- | --- | --- |
| | Pick | Look | Clean | Heat | Cool | Pick2 | All |
| HiPER-1.5B-1.5B | $\underline{98.9}_{\pm 1.9}$ | $91.7_{\pm 14.4}$ | $97.5_{\pm 4.3}$ | $\underline{90.9}_{\pm 4.7}$ | $\underline{96.7}_{\pm 2.8}$ | $\underline{91.3}_{\pm 9.6}$ | $95.3_{\pm 1.4}$ |
| HiPER-1.5B-0.5B | $97.4_{\pm 3.7}$ | $\mathbf{96.4}_{\pm 5.1}$ | $\underline{98.1}_{\pm 2.6}$ | $78.6_{\pm 6.4}$ | $85.5_{\pm 6.4}$ | $39.3_{\pm 5.1}$ | $87.5_{\pm 0.6}$ |
| HiPER-7B-7B | $\mathbf{100}_{\pm 0.0}$ | $84.8_{\pm 2.6}$ | $\mathbf{100}_{\pm 0.0}$ | $\underline{96.3}_{\pm 6.4}$ | $\mathbf{100.0}_{\pm 0.0}$ | $\mathbf{95.5}_{\pm 4.4}$ | $\mathbf{97.4}_{\pm 1.6}$ |
| HiPER-7B-1.5B | $98.6_{\pm 2.0}$ | $\underline{95.8}_{\pm 5.9}$ | $\mathbf{100}_{\pm 0.0}$ | $\mathbf{100}_{\pm 0.0}$ | $92.0_{\pm 0.0}$ | $88.9_{\pm 7.9}$ | $\underline{96.1}_{\pm 1.1}$ |

reduce large memory overhead induced by the separate critic model.

### D.3. $c_{\text{KEEP}}$ Sensitivity

*Table 4.* Sensitivity analysis on $c_{\text{KEEP}}$ for the 1.5B model on ALFWorld task.

| $c_{\text{KEEP}}$ | 0.0 | 0.1 | 0.2 | 0.3 | 0.4 | 0.5 |
| --- | --- | --- | --- | --- | --- | --- |
| Overall Succ. Rate (%) | 76.6 | 90.9 | 92.9 | 95.3 | 92.6 | 91.8 |

Table 4 shows that performance is fairly robust to the choice of the KEEP penalty $c_{\text{KEEP}}$, with a clear improvement over the no-penalty baseline. In particular, introducing a small penalty substantially boosts the overall success rate from 76.6% at $c_{\text{KEEP}} = 0$ to above 90% for all tested values in $[0.1, 0.5]$. The best result is achieved at $c_{\text{KEEP}} = 0.3$ (95.3%), while larger penalties slightly degrade performance, suggesting that moderate regularization encourages high-level exploration without overly discouraging KEEP decisions.

## E. Plan-Execute Agent Behavior

The ALFWorld trajectories illustrate effective Plan–Execute behavior. For example, in E.1, the agent first proposes the subgoal "find and cool a cup," and then KEEPs it while exploring plausible locations (countertop 1 → countertop 3 → countertop 2). After it acquires the cup ("take cup 1 from countertop 2"), it immediately SWITCHes to the next subgoal and begins the subsequent stage by going to the fridge. Overall, these trajectories show that HiPER agents can synthesize meaningful subgoals online and execute actions persistently conditioned on the current subgoal. The planning behavior is more pronounced on the ALFWorld tasks than on WebShop tasks, since WebShop episodes are significantly shorter. Nevertheless, planning is still evident—for instance, upon opening the T-shirt page, the agent revises its subgoal to prioritize "bright aqua".

### E.1. ALFWorld Trajectory 1

> **Environment (Step 1)**
>
> -= Welcome to TextWorld, ALFRED! =-
> You are in the middle of a room. Looking quickly around you, you see a cabinet 19, a cabinet 18, a cabinet 17, a cabinet 16, a cabinet 15, a cabinet 14, a cabinet 13, a cabinet 12, a cabinet 11, a cabinet 10, a cabinet 9, a cabinet 8, a cabinet 7, a cabinet 6, a cabinet 5, a cabinet 4, a cabinet 3, a cabinet 2, a cabinet 1, a coffeemachine 1, a countertop 3, a countertop 2, a countertop 1, a drawer 4, a drawer 3, a drawer 2, a drawer 1, a fridge 1, a garbagecan 1, a microwave 1, a sinkbasin 1, a stoveburner 4, a stoveburner 3, a stoveburner 2, a stoveburner 1, and a toaster 1. Your task is to: cool some cup and put it in cabinet.

> **Agent (Step 1)**
>
> <switch>SWITCH </switch><subgoal>find and cool a cup </subgoal>
> <action>go to countertop 1</action>

**Environment (Step 2)**

You arrive at countertop 1. On the countertop 1, you see a dishsponge 1, a mug 1, a potato 1, a saltshaker 1, and a tomato 1.

**Agent (Step 2)**

<switch>KEEP </switch><subgoal>find and cool a cup</subgoal>
<action>go to countertop 3</action>

**Environment (Step 3)**

You arrive at countertop 3. On the countertop 3, you see a butterknife 1, a dishsponge 3, a lettuce 2, a pot 1, a saltshaker 2, a spatula 2, and a spatula 1.

**Agent (Step 3)**

<switch>KEEP </switch><subgoal>find and cool a cup</subgoal>
<action>go to countertop 2</action>

**Environment (Step 4)**

You arrive at countertop 2. On the countertop 2, you see a bread 1, a cup 1, a fork 1, a spoon 1, and a tomato 2.

**Agent (Step 4)**

<switch>KEEP</switch><subgoal>find and cool a cup</subgoal>
<action>take cup 1 from countertop 2</action>

**Environment (Step 5)**

You pick up the cup 1 from the countertop 2.

**Agent (Step 5)**

<switch>SWITCH </switch><subgoal>cool the cup and put it in a cabinet </subgoal>
<action>go to fridge 1</action>

**Environment (Step 6)**

You arrive at fridge 1. The fridge 1 is closed.

**Agent (Step 6)**

<switch>KEEP </switch><subgoal>cool the cup and put it in a cabinet </subgoal>
<action>cool cup 1 with fridge 1</action>

**Environment (Step 7)**

You cool the cup 1 using the fridge 1.

**Agent (Step 7)**

<switch>SWITCH </switch><subgoal>put the cooled cup in cabinet </subgoal>
<action>go to cabinet 1</action>

**Environment (Step 8)**

You arrive at cabinet 1. The cabinet 1 is closed.

**Agent (Step 8)**

\<switch\>KEEP\</switch\>\<subgoal\>put the cooled cup in cabinet\</subgoal\>
\<action\>open cabinet 1\</action\>

**Environment (Step 9)**

You arrive at cabinet 1. The cabinet 1 is closed.

**Agent (Step 9)**

\<switch\>KEEP\</switch\>
\<subgoal\>put the cooled cup in cabinet\</subgoal\>
\<action\>open cabinet 1\</action\>

**Environment (Step 10)**

You open the cabinet 1. The cabinet 1 is open. In it, you see nothing.

**Agent (Step 10)**

\<switch\>KEEP \</switch\>\<subgoal\>put the cooled cup in the cabinet\</subgoal\>
\<action\>move cup 1 to cabinet 1\</action\>

## E.2. ALFWorld Trajectory 2

**Environment (Step 1)**

-= Welcome to TextWorld, ALFRED! =-
You are in the middle of a room. Looking quickly around you, you see a cabinet 4, a cabinet 3, a cabinet 2, a cabinet 1, a coffeemachine 1, a countertop 1, a diningtable 3, a diningtable 2, a diningtable 1, a drawer 1, a fridge 1, a garbagecan 1, a microwave 1, a sidetable 1, a sinkbasin 1, a stoveburner 4, a stoveburner 3, a stoveburner 2, a stoveburner 1, and a toaster 1. Your task is to: put a clean knife in sidetable.

**Agent (Step 1)**

\<switch\>SWITCH\</switch\>\<subgoal\>find a knife \</subgoal\>\<action\>go to diningtable 3\</action\>

**Environment (Step 2)**

You arrive at diningtable 3. On the diningtable 3, you see a cup 2, a lettuce 1, a peppershaker 1, a potato 3, a spatula 3, and a spoon 3.

**Agent (Step 2)**

\<switch\>KEEP\</switch\>\<subgoal\>find a knife\</subgoal\>
\<action\>go to diningtable 2\</action\>

**Environment (Step 3)**

You arrive at diningtable 2. On the diningtable 2, you see a bowl 1, a bread 1, a cup 3, a fork 3, a knife 1, and a soapbottle 2.

**Agent (Step 3)**

\<switch\>KEEP\</switch\>\<subgoal\>find a knife\</subgoal\>
\<action\>take knife 1 from diningtable 2\</action\>

**Environment (Step 4)**

You pick up the knife 1 from the diningtable 2.

**Agent (Step 4)**

<switch>SWITCH</switch><subgoal>clean the knife 1 </subgoal>
<action>go to sinkbasin 1</action>

**Environment (Step 5)**

You arrive at sinkbasin 1. On the sinkbasin 1, you see a apple 1, a butterknife 2, a potato 2, and a potato 1.

**Agent (Step 5)**

<switch>KEEP</switch><subgoal>clean the knife 1</subgoal>
<action>clean knife 1 with sinkbasin 1</action>

**Environment (Step 6)**

You clean the knife 1 using the sinkbasin 1.

**Agent (Step 6)**

<switch>SWITCH</switch><subgoal>put a clean knife in sidetable</subgoal>
<action>go to sidetable 1</action>

**Environment (Step 7)**

You arrive at sidetable 1. On the sidetable 1, you see a apple 2, a butterknife 1, a cup 1, a fork 1, a saltshaker 1, and a spoon 2.

**Agent (Step 7)**

<switch>KEEP</switch><subgoal>put a clean knife in sidetable</subgoal>
<action>move knife 1 to sidetable 1</action>

## E.3. WebShop Trajectory

**Environment (Step 1)**

Your task is to: Find me machine wash men's dress shirts with cotton spandex, classic fit, short sleeve with color: bright aqua, and size: x-small, and price lower than 60.00 dollars.Your current observation is: 'Search'.

**Agent (Step 1)**

<switch>SWITCH</switch><subgoal>Find men's dress shirts with cotton spandex, classic fit, bright aqua, size x-small, and price lower than 60.00. </subgoal><action>search[bright aqua men's dress shirts size x-small cotton spandex bright aqua price less than 60]</action>

**Environment (Step 2)**

'Back to Search'
'Page 1 (Total results: 50)'
'Next >'
'B07HRFSNL4' 'Nautica Men's Solid Crew Neck Short-Sleeve Pocket T-Shirt' '$16.05 to $40.98'
'B07F2G93BJ' 'Amazon Essentials Men's Slim-fit Long-Sleeve Solid Pocket Oxford Shirt' '$18.5'
'B09QQP3356' 'HAUKLIE Men's Sports Waffle Ribbed Polo Shirts Summer Short Sleeve Cotton Muscle Quarter-Zip Henley T-Shirt Tunics Tops' '$10.99'
'B00ED8OH2C' 'Propper Men's Pack 3 t-Shirt' '$13.37 to $47.74'
'B09KLQLLT2' 'Long Sleeve Superhero T Shirt Tank Top Mens Compression Shirt Men Workout Fitness Gym Shirt' '$19.99'
'B09PBPZ24Z' 'myhehthw Women's High Waisted Jeans for Women Distressed Ripped Jeans Slim Fit Butt Lifting Skinny Stretch Jeans Trousers' '$22.99 to $25.99'
'B09M63B87V' 'YALFJV Women Long Sleeve Crew Neck Side Button T Shirts Tunic Dress Loose Asymmetric Hem Tunic Pullover to Wear with Leggings' '$10.71 to $18.34'
'B07DKGJR74' 'Amazon Brand - Daily Ritual Women's 100% Cotton Oversized Fit V-Neck Pullover Sweater' '$20.66 to $29.2'
'B099231V35' 'Mens Casual Cargo Pants Hi Vis Viz Reflective Overalls High Visibility Safe Work Pants Outdoor Hiking Trousers Big and Tall' '$16.79'
'B07JVVDJ6L' 'Canadian Tuxedo T Shirt - Novelty Denim Tux T-Shirt' '$19.99'.

**Agent (Step 2)**

<switch>KEEP</switch><subgoal>Find men's dress shirts with cotton spandex, classic fit, bright aqua, size x-small, and price lower than 60.00.</subgoal><action>click[b07hrfsnl4]</action>

**Environment (Step 3)**

'Back to Search'
'<Prev'
'size' 'x-small' 'small' 'medium' 'large' 'large tall' 'x-large' 'x-large tall' 'xx-large' 'xx-large tall' '3x-large tall' '4x-large tall' '5x-large tall' '1x' '2x' '3x' '4x' '5x' '6x'
'color' 'navy' 'noon blue' 'grey heather' 'nautica red' 'bright white' 'true black' 'pale coral' 'cradle pink' 'bright aqua' 'mint spring' 'monaco blue' 'charcoal heather (dark)' 'barolo' 'tidal green' 'charcoal heather (light)' 'estate blue' 'patina green' 'black' 'bright aqua blue' 'bright cobalt' 'coral sands' 'deep anchor heather' 'hawaiian ocean' 'melon berry' 'pine forest green' 'shipwreck burgundy' 'white' 'zest' 'deep atlantic' 'delft' 'pale coral red'
'Nautica Men's Solid Crew Neck Short-Sleeve Pocket T-Shirt' 'Price: $16.05 to $40.98' 'Rating: N.A.'
'Description' 'Features' 'Reviews' 'Buy Now'

**Agent (Step 3)**

<switch>SWITCH</switch><subgoal>Find bright aqua men's dress shirts with cotton spandex, classic fit, size x-small, and price lower than 60.00</subgoal><action>click[bright aqua]</action>

**Environment (Step 4)**

'Back to Search'
'<Prev'
'size' 'x-small' 'small' 'medium' 'large' 'large tall' 'x-large' 'x-large tall' 'xx-large' 'xx-large tall' '3x-large tall' '4x-large tall' '5x-large tall' '1x' '2x' '3x' '4x' '5x' '6x'
'color' 'navy' 'noon blue' 'grey heather' 'nautica red' 'bright white' 'true black' 'pale coral' 'cradle pink' 'bright aqua' 'mint spring' 'monaco blue' 'charcoal heather (dark)' 'barolo' 'tidal green' 'charcoal heather (light)' 'estate blue' 'patina green' 'black' 'bright aqua blue' 'bright cobalt' 'coral sands' 'deep anchor heather' 'hawaiian ocean' 'melon berry' 'pine forest green' 'shipwreck burgundy' 'white' 'zest' 'deep atlantic' 'delft' 'pale coral red'
'Nautica Men's Solid Crew Neck Short-Sleeve Pocket T-Shirt' 'Price: $16.05 to $40.98' 'Rating: N.A.'
'Description' 'Features' 'Reviews' 'Buy Now'

**Agent (Step 4)**

<switch>KEEP</switch><subgoal>Find bright aqua men's dress shirts with cotton spandex, classic fit, size x-small, and price lower than 60.00</subgoal><action>click[x-small]</action>

**Environment (Step 5)**

'Back to Search'
'<Prev'
'size' 'x-small' 'small' 'medium' 'large' 'large tall' 'x-large' 'x-large tall' 'xx-large' 'xx-large tall' '3x-large tall' '4x-large tall' '5x-large tall' '1x' '2x' '3x' '4x' '5x' '6x'
'color' 'navy' 'noon blue' 'grey heather' 'nautica red' 'bright white' 'true black' 'pale coral' 'cradle pink' 'bright aqua' 'mint spring' 'monaco blue' 'charcoal heather (dark)' 'barolo' 'tidal green' 'charcoal heather (light)' 'estate blue' 'patina green' 'black' 'bright aqua blue' 'bright cobalt' 'coral sands' 'deep anchor heather' 'hawaiian ocean' 'melon berry' 'pine forest green' 'shipwreck burgundy' 'white' 'zest' 'deep atlantic' 'delft' 'pale coral red'
'Nautica Men's Solid Crew Neck Short-Sleeve Pocket T-Shirt' 'Price: $16.05 to $40.98' 'Rating: N.A.' 'Description' 'Features' 'Reviews' 'Buy Now'

**Agent (Step 5)**

<switch>KEEP</switch><subgoal>Find bright aqua men's dress shirts with cotton spandex, classic fit, size x-small, and price lower than 60.00</subgoal><action>click[buy now]</action>

