# OpenReview forum: "HiPER: Hierarchical Plan–Execute RL for Multi-Turn LLM Agents"
_ICML.cc/2026/Conference — ICML 2026 regular_

### Official Review · Reviewer_un1P · 2026-03-01

**Soundness:** 4
**Presentation:** 4
**Significance:** 4
**Originality:** 4
**Overall Recommendation:** 5
**Confidence:** 4

**Summary:**

The paper is in the context of RL with LLM agents in decision making, long-horizon tasks, and delayed rewards.  It starts from the premise that EL models for LLM uses flat (i.e., non hierarchical) policies. As a result the agents operate ina single time scale. the authors propose HiPER (Hierarchical Plan–Execute RL ). Hiper combines high-level planning with low-level execution. It maintains goals as an intermediate structure to decide which goals to pursue. It introduce a two-layer policy gradient for goals and for actions (as opposed to a single, flat policy).

**Compliance With Llm Reviewing Policy:**

Affirmed.

**Key Questions For Authors:**

Just the HDQN question (I consider this minor)

**Limitations:**

I don't see any such discussion but I dont consider it necessary.

**Strengths And Weaknesses:**

The motivation of the paper is clear: RL is used to improve LLM agents but they do so in a flat structure. But there is a gap when consider long-horizon settings with delayed rewards. The key observation (by others as cited by the authors) is that over long trajectories subgoals can be observed into segments of those trajectories, thereby suggesting a hierarchical structure. The key point is that HiPER makes these goals explicit in the agent's reasoning.

Hiper separates subgoaling from low-level actions. The agent dynamically decides if to pursue the current goal or switch goals. The proposed  Hierarchical Advantage Estimation (HAE) evaluates high-level decisions based on how successful was the action sequence under the subgoal and how much idid the action taken contribute towards the goal.

The paper is competent in its discussion of related work including options as semi-MDPs, and RL for LLM agents. However, I think the discussion is incomplete. For instance, the H-DQN separates goal selection and action selection. Is it becayuse HAE doesn't do reward decomposition and therefore it is not applicable in this context? do explain.

The hierarchical RL is modeled using options: a high-level option for the goal. The formulation encapsulates three decisions: whether to switch goals, if so to which goal and which action to choose. This is very clear (the brackets underneath help quite a bit). The theorem states that the gradient decomposes into these three decisions. I didn't check the proof but intuition is clear.

The empirical evaluation uses two standard and challenging benchmarks. As baselines they sue PPO, RLOO, GRPO and GiGPO, which is a solid comparision set. In the results HPER demonstrates faster learning and with less oscillations. Results show that hierarchical advantage does improve long-horizon performance.

---

> ### Author Rebuttal · Authors · 2026-03-31
>
> Thank you for your thoughtful review. We sincerely appreciate your recognition of the contributions of our work.
>
> We agree that h-DQN [1] is an important classical HRL antecedent, and we will include a more comprehensive related-work discussion in the revision. Our view is that h-DQN, and many classical HRL methods in the same family, cannot be transferred directly to the LLM-agent setting primarily because of the subgoal space. In h-DQN, the high-level controller selects goals from a structured, predefined goal space with an explicit notion of goal achievement, which in turn enables intrinsic rewards and separate high-/low-level optimization. In the LLM-agent setting, however, subgoals are open-vocabulary natural-language abstractions rather than predefined symbols, and verifying goal completion or constructing intrinsic rewards for them is non-trivial. As a result, HiPER is designed to let the agent propose subgoals directly in the open-vocabulary space, and HAE is then used to assign credit effectively to these proposed subgoals.
>
> [1] Hierarchical deep reinforcement learning: Integrating temporal abstraction and intrinsic motivation. Neurips 2016.

---

> > ### Author Rebuttal · Reviewer_un1P · 2026-04-01
> >
> > Authors clarified relation versus H-DQN.

---

> > > ### Author Response · Authors · 2026-04-07
> > >
> > > We sincerely appreciate your efforts in reviewing our paper and your recognition of our contributions. We will incorporate the additional discussion versus h-DQN in our revision.

---

### Official Review · Reviewer_WgWB · 2026-03-02

**Soundness:** 3
**Presentation:** 3
**Significance:** 3
**Originality:** 2
**Overall Recommendation:** 3
**Confidence:** 3

**Summary:**

This paper introduces HiPER, a hierarchical reinforcement learning framework for training large language model (LLM) agents in multi-turn, long-horizon, sparse-reward environments. The core contributions are: (1) an explicit Plan–Execute interface that decomposes agent behavior into high-level subgoal planning and low-level action execution; (2) a hierarchical advantage estimation scheme (HAE) assigning credit both within and across subgoal segments to improve credit assignment; and (3) theoretical analysis justifying unbiasedness and variance reduction of the proposed estimator versus flat RL baselines. Experimental results on ALFWorld and WebShop benchmarks demonstrate substantial improvements in success rate, sample efficiency, and training stability, especially on complex tasks with multiple dependent subtasks.

**Compliance With Llm Reviewing Policy:**

Affirmed.

**Final Justification:**

The rebuttal clarified several points and strengthened the paper in some respects, especially by adding further comparisons, ablations, and broader empirical evidence. These additions make the method more convincing. However, my main concerns remain only partially resolved: the empirical evaluation is still not comprehensive enough, and the paper’s originality remains somewhat limited relative to prior hierarchical RL ideas. Overall, the rebuttal improves the paper, but not enough to change my final assessment.

**Key Questions For Authors:**

### 1. Generalization beyond current benchmarks

The empirical evaluation is limited to **ALFWorld and WebShop**. How does HiPER perform on substantially different settings, such as **multi-hop QA, retrieval-augmented reasoning, or longer-horizon tool-use benchmarks**? Are the hierarchical gains consistent across **different LLM backbones**?

---

### 2. Positioning relative to recent hierarchical RL methods (2024–2025)

Several closely related **hierarchical RL methods for LLM agents (2024–2025)** are not included as baselines or in Table 1. What was the rationale for excluding these approaches?

Without **direct comparison to contemporary hierarchical competitors**, it is difficult to assess whether HiPER truly advances the state of the art. Clarifying the **key algorithmic distinctions** and, ideally, adding explicit benchmarks against these methods would substantially strengthen the paper.

---

### 3. Hierarchical design and experimental breadth

The method fixes a **two-level hierarchy with binary switch/keep decisions**. How sensitive are results to this architectural choice? Have **deeper hierarchies or alternative segmentation mechanisms** been explored?

Broader **environment diversity** and ablations over **hierarchical configurations** would improve confidence in the robustness and generality of the approach.

**Limitations:**

The current evaluation is limited to a small set of benchmark environments, lacking validation on more diverse experimental platforms. This restricts the strength of the generalization claims. In addition, while the theoretical analysis is rigorous, it primarily builds upon existing hierarchical RL principles and does not clearly establish fundamentally new theoretical contributions. Finally, the hierarchical design is restricted to a fixed two-level structure with binary switch/keep decisions, without exploring richer or more flexible hierarchy formulations.

**Strengths And Weaknesses:**

### Strengths

**1. Well-motivated hierarchical design.**
The Plan–Execute interface is principled and conceptually clean. It elevates trajectory segmentation—often handled implicitly in LLM agents—into an explicit and trainable structure. Figure 1 effectively illustrates the contrast between flat and hierarchical agents, clearly conveying the intuition and practical relevance of the approach.

**2. Theoretical clarity and rigor.**
The derivation of the hierarchical policy gradient and the variance reduction analysis of the advantage estimator are clearly presented and formally justified. The proofs are detailed and appear sound. While not radically new from a statistical standpoint, the theory provides a coherent formalization tailored to the hierarchical LLM-agent setting.

**3. Strong empirical performance.**
Across environments and model scales, HiPER consistently outperforms competitive RL baselines. The improvements are especially pronounced in tasks requiring multi-step subgoal completion (e.g., Pick2 and Look in ALFWorld), supporting the claim that explicit hierarchy benefits long-horizon reasoning and planning.

---

### Weaknesses

**1. Limited theoretical novelty.**
Although the theoretical development is rigorous, much of it adapts established hierarchical RL and variance-reduction principles to the LLM-agent setting rather than introducing fundamentally new statistical or optimization results. The paper would benefit from more clearly articulating which aspects constitute genuine theoretical advances versus extensions of known results.

**2. Incomplete coverage of recent related work.**
The discussion and experimental comparison with recent hierarchical RL methods for LLM agents is insufficient. Several contemporary approaches on structured planning, option learning, or hierarchical policy optimization in LLM-based agents are not discussed or empirically compared. This omission makes it difficult to fully assess HiPER’s distinct advantages or limitations relative to the latest literature.

**3. Limited empirical generalization scope.**
Experiments are restricted to ALFWorld and WebShop. While these are challenging benchmarks, they do not cover the full spectrum of hierarchical multi-turn RL settings for LLMs. The absence of validation on additional or more recent benchmarks limits claims about general applicability.

**4. Lack of ablations on hierarchy design.**
The hierarchy is fixed to two levels with binary switch/keep decisions. The paper does not explore:

* Deeper hierarchies (more than two levels),
* More granular option structures,
* Automatic segmentation boundary discovery mechanisms.

Without these ablations, it is unclear whether the chosen hierarchical structure is optimal or merely sufficient.

---

> ### Author Rebuttal · Authors · 2026-03-31
>
> Thank you for your review. Below are our responses point by point.
>
> **1. Theoretical Novelty.**
> We wish to clarify that this paper is not really a theoretical work, and our goal is not to introduce fundamentally new theoretical results. Rather, we focus on solving the challenge of long-horizon credit assignment in LLM agents. While our approach builds upon established hierarchical RL and variance reduction principles, instantiating these concepts to the autoregressive, language-based action space of LLM agents is non-trivial. Our core theoretical contribution is the formal derivation of a new HRL algorithm tailored to this setting. We also believe that our proposed HAE is of broader interest to the classical HRL research. It provides a unified way to learn under the options framework without handcrafted heuristics. As preliminary evidence, we include results on classical RL environments (FourRooms, HalfCheetah):
> https://anonymous.4open.science/r/icml2026_722F/hrl.png
>
> HAE can outperform strong HRL baselines such as DAC [1] and AHP [2], suggesting its potential value for option learning more generally.
>
> **2. Related Work.**
> The review does not specify which 2024–2025 papers it refers to, and we would be glad to discuss specific papers the reviewer suggests. Meanwhile, we additionally identified ADaPT [3], HiPlan [4], GLIDER [5], and ArCHer [6] as related work. ADaPT and HiPlan are prompt-based planning without RL training. GLIDER and ArCHer are **offline** hierarchical RL methods for LLM agents, relying on sentence-level critics trained via offline/off-policy RL and token-level actors. HiPER differs fundamentally from these methods. It is online, on-policy hierarchical RL that instantiates hierarchy with a single autoregressive LLM model, with HAE assigning credit over subgoal segments and switching boundaries, rather than over sentence/token levels. The benefit of HiPER is that it combines a simple unified actor with HAE, which effectively improves long-horizon credit assignment. Empirically, this translates into stronger performance, as shown below.
>
> **Prompt-based planning**
> Method|Type|Model|ALFWorld SR%|WebShop SR%|
> -|-|-|-|-|
> ADaPT|Prompt-based|GPT3.5 (175B)|71.6|44.0|
> HiPlan|Prompt-based|Llama3.3-70B|94.0|40.0|
> HiPER|Online hierarchical RL|Qwen2.5-1.5B|95.3|71.4|
> HiPER|Online hierarchical RL|Qwen2.5-7B|97.4|83.3|
>
> Prompt-based methods, despite using much larger base models, underperform HiPER on both tasks, suggesting that prompt planning alone is insufficient to substantially improve performance.
>
> **Offline Hierarchical RL baseline**
> Method|Type|Model|ALFWorld SR%|
> -|-|-|-|
> GLIDER|SFT+Offline hierarchical RL|Llama-3-8B|75.4|
> HiPER|Online hierarchical RL|Llama-3-8B|89.8|
>
> For fair comparison with GLIDER and to show generality across models, **we evaluated HiPER with Llama-3-8B on ALFWorld**. HiPER outperforms GLIDER by 14.4%, highlighting the value of online exploration and HAE in long-horizon interactive settings.
>
> **3. More Benchmarks.**
> We provide additional results for HiPER in a Vision-Language environment.
>
> **VisualSokoban (Qwen 2.5-VL-3B)**
> Method|SR%|
> -|-|
> HiPER|90.6|
> GRPO|67.1|
> GiGPO|81.0|
>
> On VisualSokoban, HiPER achieves 90.6% success rate, outperforming GRPO and GiGPO, showing HiPER's generalizability to diverse models and tasks.
>
> **4. Hierarchy.**
> We use a two-level hierarchy since it is the simplest structure that already matches the long-horizon behaviors we observe in practice. Extending to deeper hierarchies requires substantial algorithmic changes, including extra levels of decisions and cross-level credit assignment. This can be an important future direction, but currently we believe it is beyond the scope of this paper. We also clarify that boundary segmentation is already automatic: at each turn, the agent decides by itself whether to KEEP or SWITCH the current subgoal.
>
> We agree that more ablation studies could further strengthen the paper, and provide additional results below.
>
> **Ablations**
> Method|ALFWorld SR%|Avg. Segment Len.|
> -|-|-|
> Plan-Execute+flat PPO|81.3|~1|
> HiPER(no switch adv.)|79.7|~1|
> HiPER(no high-value)|77.3|2.77|
> HiPER|95.3|2.68|
>
> For no switch adv., we replace the switching advantage by the high-level advantage, and for no high-value, we remove the high-level value head and use MC returns to calculate high-level advantage. These results show that removing each core component will result in performance degradation: Removing high-level value amplifies variance (see https://anonymous.4open.science/r/icml2026_722F/adv-std.png), and removing switching advantage results in switching at almost every turn, leading to average subgoal segment length of 1.
>
> [1] DAC: The double actor-critic architecture... 2019
>
> [2] Unified inter and intra options... 2011
>
> [3] Adapt: As-needed decomposition and... 2024
>
> [4] Hiplan: Hierarchical planning for llm-based... 2025
>
> [5] Divide and Conquer: Grounding LLMs as Efficient... 2025
>
> [6] ArCHer: Training Language Model... 2024

---

> > ### Author Rebuttal · Reviewer_WgWB · 2026-04-04
> >
> > Thank you for the authors’ response. However, the newly added comparison methods still do not constitute a sufficiently comprehensive evaluation. Therefore, I will maintain my original score.

---

### Official Review · Reviewer_vGge · 2026-03-11

**Soundness:** 3
**Presentation:** 3
**Significance:** 3
**Originality:** 3
**Overall Recommendation:** 5
**Confidence:** 4

**Summary:**

This paper proposes HiPER, which addresses the challenge of training LLM agents for long-horizon, sparse-reward tasks where flat reinforcement learning struggles with credit assignment across extended action sequences. HiPER is a hierarchical RL framework that makes explicit subgoals observed in agent trajectories, allowing actions to be naturally organized into segments persisting across multiple turns. Based on this, the paper introduces Hierarchical Advantage Estimation (HAE), a two-timescale advantage estimator that assigns credit both within subgoal segments and across segment boundaries, providing more effective learning signals than flat GAE. Empirically, HiPER demonstrates improved performance on several long-horizon tasks and converges much faster than standard RL training.

**Compliance With Llm Reviewing Policy:**

Affirmed.

**Final Justification:**

My concerns are mostly addressed. I will maintain my positive score toward acceptance.

**Key Questions For Authors:**

1. It seems that when there is planning (<switch>xxx), there is no <think> tag. Is this a careful design?
2. From Figure 2, it seems that both PPO and GRPO have not converged yet. Is my understanding correct that the results tabulated in table 1 is provided by models trained with a time budget (probably for the purpose of fair comparison)? If given more time, would the baseline models perform better?
3. It seems that there are only <subgoal> xml tags but no <plan> tags. Do the authors think that there is no need to provide a high-level plan before entering the actual subgoals?

**Limitations:**

While the paper analyzes switching behavior and provides ablation studies, a dedicated discussion of when and why HiPER might fail (i.e., qualitative analysis) would strengthen the paper's honesty about its limitations.

**Strengths And Weaknesses:**

Strength
- The paper proposes an interesting method that hardcodes planning as part of the model's reasoning process and also allows for dynamically updating the planning via an R1-style XML syntax.
- Based on the proposed method, the authors provide an innovative RL algorithm to support this type of training, with pretty reasonable theoretical backup and promising experimental results.
- Such a structured reasoning process also potentially provides convenience for context management, which can group the context by its segments easily, allowing for wider impact.

Weakness
- There lacks a comparison with a pure harness-based approach. Many agent harness nowadays supports making and updating plans during rollout via prompting (e.g., opencode and claude code). For a more rigorous comparison, the paper can compare with these training-free approaches to find the gap.
- There is no qualitative analysis that demonstrates some of the common failure modes of this plan-execute reasoning process.

---

> ### Author Rebuttal · Authors · 2026-03-31
>
> Thank you for your thoughtful review. We sincerely appreciate your recognition of the contributions of our work. Below we provide response to your questions, point by point.
>
> **1. Comparison with prompt-based methods.**
> We include additional comparison with two prompt-based planning methods (ADaPT [1] and HiPlan [2]) in the table below.
>
> **Comparison with prompt-based planning methods**
> | Method | Type | Base model | ALFWorld SR% | WebShop SR% |
> |---|---|---|---|---|
> | **ADaPT** | Prompt-based planning | GPT-3.5 (175B) | 71.6 |44.0 |
> | **HiPlan** | Prompt-based planning | Llama3.3-70B | 94.0 | 40.0 |
> | **HiPER** | Online hierarchical RL | Qwen2.5-1.5B | 95.3 | 71.4 |
> | **HiPER** | Online hierarchical RL | Qwen2.5-7B | **97.4** | **83.3** |
>
> Prompt-level planning methods, despite using base models with much larger size than Qwen2.5 used in HiPER, cannot match the performance of HiPER in both tasks. These results highlight that prompt-level planning alone is not sufficient for effectively improving LLM agentic performance.
>
> **2. Qualitative Analysis**
> Thank you for the suggestion. We provide representative HiPER failure examples in the anonymous link:
> https://anonymous.4open.science/r/icml2026_722F/fail-example.png
>
> In failed trajectories, the proposed subgoals often collapse either to the full task description or to a single primitive action. These low-quality subgoals become much less frequent as training progresses, but they are not fully eliminated. Ensuring consistently high-quality subgoal proposals remains to be explored for future work. See the following link for a quantitative summary of subgoal quality:
> https://anonymous.4open.science/r/icml2026_722F/goal-tab.png
>
> We will also add this discussion of HiPER's potential limitations in the revision.
>
> **3. Baselines not converged yet.**
> Thank you for the sharp observation. You are right that Table 1 results are provided by models trained with a fixed 150-iteration budget, and we agree that PPO/GRPO do not appear fully converged at that point from Figure 2. To assess whether the baselines would improve with more training, we further trained HiPER and GRPO on ALFWorld with Qwen2.5-1.5B up to 250 iterations, and we show the curves in the anonymous link:
> https://anonymous.4open.science/r/icml2026_722F/long-iter.png
>
> GRPO performance largely plateaus after 180 iterations, with the best success rate of 86.3% at iteration 180. HiPER's best performance is at iteration 245, with success rate 96.5%. This suggests that HiPER’s advantage is not only in sample efficiency, but also in final attainable performance under longer training.
>
> **4. Questions 1 and 3.**
> We are unfortunately not fully sure we understand the notation in the comment, since the notations appear as “[object Object]” on our side. Could you please clarify which specific tags or fields you are referring to? We would appreciate clarification before responding in more detail.
>
> [1] Adapt: As-needed decomposition and planning with language models. NAACL Findings 2024
>
> [2] Hiplan: Hierarchical planning for llm-based agents with adaptive global-local guidance. arXiv preprint 2025

---

> > ### Author Rebuttal · Reviewer_vGge · 2026-04-01
> >
> > Please refer to my resposne to the authors' rebuttal.

---

> > > ### Author Response · Authors · 2026-04-08
> > >
> > > Dear Reviewer vGge,
> > >
> > > We sincerely appreciate your effort in reviewing our work and your acknowledgment of our contributions. We noticed that you have follow-up questions for us, however, we are unable to view your response to the rebuttal on our side. If you could please edit the rebuttal acknowledgement or final justification so we can see the questions, we will try our best to answer them by editing this reply. If unfortunately the rebuttal period ends before we are able to reply, we are also happy to reflect the additional discussion in the revised manuscript. Thank you!

---

### Official Review · Reviewer_HB58 · 2026-03-12

**Soundness:** 2
**Presentation:** 3
**Significance:** 2
**Originality:** 2
**Overall Recommendation:** 4
**Confidence:** 3

**Summary:**

This paper argues that flat, step-wise RL training for multi-turn LLM agents is inefficient and unstable in long-horizon, sparse/delayed-reward environments, because it fails to explicitly represent the subgoal-level structure that successful trajectories naturally exhibit. It proposes HiPER, which enforces a Plan-Execute interface where the LLM outputs at each turn a KEEP/SWITCH decision, a subgoal, and an action, turning trajectories into subgoal-consistent segments with explicit switching boundaries. It introduces Hierarchical Advantage Estimation (HAE), which uses within-segment low-level advantages and boundary-level high/switching advantages to train action execution and subgoal switching. The paper provides a theory that the resulting policy-gradient estimator is unbiased up to standard GAE bootstrapping and critic approximation errors and can reduce variance versus flat GAE under simplifying assumptions. Experiments show higher performance, with especially strong gains on multi-subtask categories.

**Compliance With Llm Reviewing Policy:**

Affirmed.

**Final Justification:**

My initial concerns included baseline fairness, switching/advantage mechanism validation, implementation clarity, missing ablations, and limited evidence for robustness and generality. Across two rounds of rebuttal, the authors effectively addressed most of these issues, although I still have some uncertainty about the appropriateness of the "hierarchy" characterization, since both subgoal-level and action-level decisions are realized within a single autoregressive model.

Overall, the authors’ responses resolved the majority of my concerns, so I have raised my score.

**Key Questions For Authors:**

See Weaknesses

**Limitations:**

Yes

**Strengths And Weaknesses:**

**Strengths**

-	This paper is well motivated.
-	The proposed approach is conceptually simple and broadly applicable.

**Weaknesses**

-	In Section 5, the set of baselines is largely limited to algorithmic variants, which may not constitute a fully fair comparison. Simply changing the training algorithm does not control for other major design choices that affect long-horizon agent performance. For a stronger evaluation, the authors should also compare against recent hierarchical agent architectures, prompt-based planning/segmentation methods, and non-RL alternatives such as supervised or preference-based fine-tuning (where applicable), to better substantiate the claimed advantages of the proposed hierarchical approach.
-	In Section 4.3, the switching signal is defined through a difference between value estimates (high-level vs. option-conditioned low-level values), which may be sensitive to critic miscalibration and could lead to premature or delayed subgoal switching; a more explicit analysis of failure cases (e.g., incorrect switching events) and critic-quality sensitivity would strengthen the claims.
- In Section 3, while the switching, high-level, and low-level policies are introduced as separately parameterized components, the paper later states that they are realized by a single autoregressive LLM policy $\pi_{\theta}$. Clarifying whether the actor parameters are fully shared in practice (and, if so, how token-level log-probabilities are partitioned across $\texttt{\langle switch\rangle}$, $\texttt{\langle subgoal\rangle}$, and $\texttt{\langle action\rangle}$ for PPO-style updates) would improve readability and reproducibility.
-	In Section 4.3, while the paper motivates HAE as improving signal quality via boundary-aware bootstrapping, it does not directly validate this mechanism empirically (e.g., by measuring advantage variance/SNR or demonstrating that boundary bootstrapping reduces long-horizon credit noise relative to flat GAE).
-	In Section 5, the approach relies on open-vocabulary subgoals learned from outcome-only rewards, yet the paper provides limited quantitative diagnostics on subgoal quality (e.g., length, redundancy, semantic drift/collapse) beyond qualitative trajectories, making it harder to assess robustness and interpretability of the learned hierarchy.
-	In Section 5, the reviewer wonders whether the gains attributed to the hierarchical design remain robust under component-wise ablations. Specifically, could the authors report results for (1) Plan–Execute prompting only trained with a flat PPO/GAE estimator, (2) removing the switching objective/advantage (or enforcing always-KEEP / fixed-period SWITCH), and (3) removing the high-level value head (or replacing $V^{high}$ with a simpler baseline)? In addition to success rates, please report switch frequency / average segment length and basic subgoal statistics (e.g., length and redundancy) to more clearly quantify the contribution of hierarchy.
-	In Section 5, the benchmarks are largely restricted to ALFWorld and WebShop, and the experiments are limited to Qwen2.5 (1.5B/7B). To better support the claim that the approach is “broadly applicable,” it would be important to replicate the results across more diverse LLM families (e.g., Llama or Mistral) and additional agent domains (e.g., tool-use settings, longer-horizon tasks, and more strongly partially observable environments).
-	The reviewer wonders about additional case studies to clarify when and why HiPER helps: (1) representative failure cases (e.g., premature/late switching, subgoal drift/collapse), and (2) side-by-side trajectories showing how HiPER recovers from errors that strong baselines fail to overcome. Concretely, please include a few paired examples with the same initial state/instruction, highlighting the first divergence point, the resulting subgoal/segment structure, and whether recovery was possible.

---

> ### Author Rebuttal · Authors · 2026-03-31
>
> We appreciate your thoughtful review. Below are our responses point by point.
>
> **1. More baselines.**
> Our original comparisons focused on controlled online RL in the same interactive LLM-agent setting. We also already account for prompt effects: Appendix D.2 shows that Plan-Execute alone improves flat RL, but still falls short of full HiPER. We agree broader baselines are useful, and we add more baselines: ADaPT [1], HiPlan [2], which are prompt-based planning methods, and GLIDER [3], which is an offline hierarchical RL method.
>
> **Prompt-based planning**
> |Method|Type|Model|ALFWorld SR%|WebShop SR%|
> |-|-|-|-|-|
> |ADaPT|Prompt-based|GPT3.5 (175B)|71.6|44.0|
> |HiPlan|Prompt-based|Llama3.3-70B|94.0|40.0|
> |HiPER|Online hierarchical RL|Qwen2.5-1.5B|95.3|71.4|
> |HiPER|Online hierarchical RL|Qwen2.5-7B|97.4|83.3|
>
> Prompt-based methods, despite using much larger base models than HiPER, still underperform HiPER on both tasks, suggesting that prompt planning alone is insufficient to substantially improve agent performance.
>
> **Offline Hierarchical RL baseline**
> Method|Type|Model|ALFWorld SR%|
> -|-|-|-|
> GLIDER|SFT+Offline hierarchical RL|Llama-3-8B|75.4|
> HiPER|Online hierarchical RL|Llama-3-8B|89.8|
>
> For fair comparison with GLIDER and to show generality across models, **we evaluated HiPER with Llama-3-8B on ALFWorld**. HiPER outperforms GLIDER by 14.4%, highlighting the value of online exploration and HAE in long-horizon interactive settings.
>
> **2. Qualitatitve analysis.**
> We include a representative comparison with GiGPO here: https://anonymous.4open.science/r/icml2026_722F/example.png
>
> **3. Critic sensitivity.**
> We evaluated HiPER’s sensitivity to critic quality by using critic models smaller than the actor.
> Method(Actor size-Critic size)|ALFWorld SR%|
> -|-|
> HiPER(1.5B-1.5B)|95.3|
> HiPER(1.5B-0.5B)|87.5|
> GiGPO (1.5B)|86.7|
> -|-|
> HiPER (7B-7B)|97.4|
> HiPER (7B-1.5B)|96.1|
> GiGPO (7B)|90.8|
>
> With smaller critics, HiPER remains competitive and outperforms the strong flat-RL baseline GiGPO, suggesting HiPER is fairly robust to critic quality.
>
> **4. Clarification.**
> The actor parameters are fully shared: switching, subgoal, and action policies are realized by a single autoregressive LLM. The model generates $\langle \texttt{switch} \rangle$, $\langle \texttt{subgoal} \rangle$, and $\langle \texttt{action} \rangle$ sequentially, so later fields are naturally conditioned on earlier ones. We partition the output into the three fields and sum token-level log-probabilities within each field to obtain the field-level log-probabilities used later. We will clarify this in Appendix B.
>
> **5. Advantage Variance.**
> We track batch-level advantage standard deviation over training steps for HiPER, flat PPO, and HiPER without high-level value head. HiPER has **57.3% lower** average advantage std compared to flat PPO, while removing the high-level value head amplifies variance, increasing std by 30% over flat PPO.
> Advantage std curves are provided here:
> https://anonymous.4open.science/r/icml2026_722F/adv-std.png
> The results support that HAE improves credit assignment by reducing long-horizon credit noise, while naive hierarchy without proper value separation can amplify it.
>
> **6. Quantitative analysis of subgoal.**
> We compare one batch of trajectories from the base model before training with one from the trained HiPER agent using four metrics: format correctness, irrelevant subgoals, single-action collapse, and full-task collapse.
> The full table is provided here:
> https://anonymous.4open.science/r/icml2026_722F/goal-tab.png
> Overall, the trained agent produces better subgoals, with lower irrelevant (6.4% vs. 27.2%) and collapse rates (15.9% vs. 35.4% combined).
>
> **7. Ablations and benchmarks.**
> We provide additional ablation studies on HiPER, and an additional environment VisualSokoban.
>
> **Ablations**
> Method|ALFWorld SR%|Avg. Segment Len|
> -|-|-|
> Plan-Execute+flat PPO|81.3|~1|
> HiPER(no switch adv.)|79.7|~1|
> HiPER(no high-value)|77.3|2.77|
> HiPER|95.3|2.68|
>
> For no switch adv., we replace the switching advantage by the high-level advantage, and for no high-value, we remove the high-level value head and use MC returns to calculate high-level advantage. These results show that removing each core component will result in performance degradation for HiPER: Removing high-level value results in amplified variance, while removing switching advantage results in degenerate switching (switching at almost every turn, leading to average subgoal segment length of 1).
>
> **VisualSokoban (Qwen 2.5-VL-3B)**
> Method|SR%|
> -|-|
> HiPER|90.6|
> GRPO|67.1|
> GiGPO|81.0|
>
> On VisualSokoban, HiPER achieves 90.6% success rate, outperforming GRPO and GiGPO, demonstrating HiPER's generalizability to Vision-Language models and tasks.
>
> [1] Adapt: As-needed decomposition and... NAACL Findings 2024
>
> [2] Hiplan: Hierarchical planning for llm-based... arXiv preprint 2025
>
> [3] Divide and Conquer: Grounding LLMs... ICML 2025

---

> > ### Author Rebuttal · Reviewer_HB58 · 2026-04-02
> >
> > I thank the authors for their rebuttal. The additional experiments and discussions have mostly resolved all of my concerns. Based on the improvements made in the rebuttal, I am raising my score.

---

> > > ### Author Response · Authors · 2026-04-07
> > >
> > > Thank you for your follow-up questions. Below we provide response to your concerns.
> > >
> > > **Same backbone.**
> > >
> > > We believe the previous results already suggest HiPER's advantage, since **prompt methods with much larger backbones are outperformed by HiPER**, and **GLIDER is compared with HiPER on the exact same Llama-3-8B backbone**. Nevertheless, to provide a more controlled comparison, we add results of **ADaPT and GLIDER on matched Qwen backbones**. For HiPlan, we were unable to provide additional results other than reported in their paper, since the authors did not release code. For GLIDER, we were unable to evaluate WebShop because the released implementation requires additional hierarchically annotated data that is not provided by the authors.
> > >
> > > |Method|Model|ALFWorld SR%|WebShop SR%|
> > > |-|-|-|-|
> > > |ADaPT|Qwen2.5-1.5B|7.14|14.8|
> > > |ADaPT|Qwen2.5-7B|21.4|19.5|
> > > |GLIDER|Qwen2.5-1.5B|32.1|-|
> > > |GLIDER|Qwen2.5-7B|56.6|-|
> > > |HiPER|Qwen2.5-1.5B|95.3|71.4|
> > > |HiPER|Qwen2.5-7B|97.4|83.3|
> > >
> > > These matched-backbone results further support our conclusion: **prompt-based planning alone is insufficient, and GLIDER also remains substantially below HiPER in the interactive setting**.
> > >
> > > **More Switching Analysis and Critic Sensitivity.**
> > >
> > > We now analyze representative switching failure modes (premature and delayed) in HiPER and include more statistical results. Concretely, **premature switching** refers to the agent switches subgoals before the current one has been completed, and **delayed switching** refers to the agent continues pursuing a subgoal that has already been completed or has failed. We briefly describe one example observed for each case:
> > >
> > > **Premature switching.** For a task "clean a soapbar and put it on table", the agent first takes the soapbar, but fails to clean it because it outputs an invalid action (clean soapbar 1) instead of the correct sequence (go to sinkbasin 1, clean soapbar 1 with sinkbasin 1). Then it switches prematurely to the subgoal of putting the soapbar on the table, when the cleaning stage is unfinished. However, the agent later switches back, successfully cleans the soapbar, and eventually completes the task.
> > >
> > > **Delayed switching.** For a task “find two cups and put them in microwave”, after successfully putting one cup in the microwave, the agent fails to switch away from the current placement subgoal. Instead, it keeps pursuing the completed subgoal, and repeatedly takes the second cup in and out of the microwave, causing the episode to fail.
> > >
> > > Note that incorrect switching does not necessarily cause task failure. To provide a rough sense of the association, 45% of episodes with premature switching failed, and 69% of episodes with delayed switching failed.
> > >
> > > We use LLM-as-Judge to determine premature/delayed switching in each episode. Below we provide a comprehensive summary of switching-related statistics (including **both switch frequency and segment length**) at different stages of training:
> > >
> > > **1.5B Actor-1.5B Critic**
> > > |Stage|Switch Freq. %|Segment Len.|Premature %|Delayed %|SR%|
> > > |-|-|-|-|-|-|
> > > |Early|96.8|1.03|41.6|46.9|8.6|
> > > |Mid|58.9|1.73|13.2|17.5|71.4|
> > > |Late|36.7|2.68|9.3|6.4|95.3|
> > >
> > > **1.5B Actor-0.5B Critic**
> > > |Stage|Switch Freq. %|Segment Len.|Premature %|Delayed %|SR%|
> > > |-|-|-|-|-|-|
> > > |Early|95.1|1.05|41.4|53.1|4.7|
> > > |Mid|56.4|1.75|25.0|17.9|68.8|
> > > |Late|39.7|2.52|14.1|13.3|87.5|
> > >
> > > We observe that **incorrect switching becomes much less frequent** as training progresses, suggesting that HiPER gradually learns a more accurate switching policy. Compared with the 1.5B critic, the 0.5B critic setting yields higher rates of incorrect switching, along with lower success rate. This suggests that **degraded critic does affect the switch timing quality, and may be a contributor to the performance drop**.
> > >
> > > **Hierarchy.**
> > >
> > > We would like to clarify that our notion of **hierarchy is defined on the level of decision-making**, not on the model architecture. Concretely, subgoal decisions are temporally extended over multiple turns and are updated by explicit switch decisions, where execution actions are generated every turn. Using a single LLM to realize these policies is not only a natural **implementation choice**, but also a key **practical advantage** of HiPER, rather than evidence against the presence of hierarchy. First, the autoregressive nature of LLMs naturally provides the **desired conditioning of policies**, since the switch, subgoal, and action fields are generated sequentially. Second, a structurally separated hierarchy would require **changes in the LLM architecture** and substantial additional machinery for representing, passing, and optimizing subgoals across levels. This would make the method significantly **harder to integrate into existing LLM-agent systems**, thereby undermining its practicality. To give an intuitive analogy, even in human learning, hierarchical behavior is often reflected in the way decisions are organized across time scales, rather than in explicit architectural separation.

---

### Decision · Program_Chairs · 2026-04-30

**Decision:**

Accept (regular)

**Comment:**

This paper proposes a novel hierarchical reinforcement learning strategy for LLMs. The reviewers generally found the approach to be compelling, with strong experimental results. There were some concerns about the limited scope of the experiments (e.g., limited benchmarks, baselines, models, etc.), these were addressed to some degree in the rebuttal though there is room for further improvement. On the whole, the reviewers agree that this paper makes a good contribution.